# The $\mu$Dose-system: determination of environmental dose rates by combined alpha and beta counting – performance tests and practical experiences

Thomas Kolb[1], Konrad Tudyka[2], Annette Kadereit[3], Johanna Lomax[1], Grzegorz Poręba[2], Anja Zander[4], Lars Zipf[5], and Markus Fuchs[1]

[1]Justus Liebig University Giessen, Department of Geography, Senckenbergstr. 1, D-35390 Giessen, Germany
[2]Silesian University of Technology, Institute of Physics - Centre for Science and Education, Division of Geochronology and Environmental Isotopes, ul. S. Konarskiego 22B, 44-100 Gliwice, Poland
[3]University of Heidelberg, Geographical Institute, Im Neuenheimer Feld 348, D-69120 Heidelberg, Germany
[4]University of Cologne, Institute of Geography, Otto-Fischer-Straße 4, D-50674 Cologne, Germany
[5]Université Libre de Bruxelles, Laboratoire de Glaciologie (GLACIOL), Belgium

**Correspondence:** Thomas Kolb (thomas.r.kolb@geogr.uni-giessen.de)

**Abstract.** The $\mu$Dose-system is a recently developed analytical instrument applying a combined $\alpha$- and $\beta$-sensitive scintillation technique for determining the radioactivity arising from the decay chains of $^{235}$U, $^{238}$U and $^{232}$Th as well as from the decay of $^{40}$K. The device was designed to meet the particular requirements of trapped charge dating methods and allows the assessment of environmental (i.e. low) levels of natural radionuclides. The $\mu$Dose-system was developed as a low-cost laboratory equipment, but a systematic test of its performance is still pending. For the first time, we present results from a comprehensive performance test based on an inter-laboratory comparison. We compare the results gained with $\mu$Dose-measurements with those from thick source alpha counting (TSAC), inductively coupled plasma optical emission spectrometry (ICP-OES) and low-level high-resolution gamma spectrometry (HRGS) applied in five participating laboratories. In addition, the reproducibility and accuracy of $\mu$Dose-measurements were tested on certified reference materials distributed by the International Atomic Energy Agency (IAEA; RGU-1, RGTh-1 and RGK-1) and on two loess standards (Nussy and Volkegem) frequently used in trapped charge dating studies. We compare $\mu$Dose-based results for a total of 47 sediment samples with results previously obtained for these materials by well-established methods of dose rate determination. The investigated natural samples cover a great variety of environments, including fluvial, aeolian, littoral, colluvial and (geo-)archaeological sites originating from high- and low-mountain regions as well as from lowlands in tropical areas, drylands and mid-latitude zones of Europe, Africa, Australia, Central Asia and the Americas. Our results suggest the $\mu$Dose-system's capability of assessing low-level radionuclide contents with very good accuracy and precision comparable to well-established dosimetry methods. Based on the results of our comparative study and with respect to the practical experiences gained so far, the $\mu$Dose-system appears to be a promising tool for trapped charge dating studies.

# 1 Introduction

Over the last two decades, trapped charge dating techniques have become commonly applied standard tools for age determination of sediments in palaeo-environmental and geo-archaeological research. The vast arsenal of luminescence and electron spin resonance (ESR) dating methods (e.g., Bateman, 2019; Grün, 1989; Preusser et al., 2008) allows the direct dating of sedimentation processes, heating events, and for ESR the precipitation of minerals. Ages gained with trapped charge dating are derived from doses (energy per mass unit), stored by minerals such as quartz and feldspars, which are ubiquitously present in natural sediments and other materials such as tooth enamels and ceramics. These minerals may therefore be used as dosimeters. The dating events are associated with processes which involve the energetic stimulation of these minerals either by sunlight exposure (e.g., during sediment transport) or by natural or artificial heating (e.g., rocks fritted during volcanic eruptions; ceramics heated in kilns). The optical or thermal stimulation releases the dose previously stored within the crystal lattices of the involved dosimeters thus "zeroing" the "luminescence clock" (e.g., Bateman, 2019; Wagner, 1998). When the minerals are no longer stimulated (e.g. after sediment deposition or after the end of the heating event), they remain exposed to the natural ionizing radiation arising from both cosmic radiation and the radioactive decay of members of the $^{238}$U, $^{235}$U and $^{232}$Th decay chains as well as from the decay of $^{40}$K in the surrounding sediments. This ongoing exposure to ionizing radiation results in a time-dependent accumulation of radiation doses within the minerals (e.g., Preusser et al., 2008). The total amount of dose absorbed under natural conditions since the last stimulation event is termed the palaeodose and can be determined in the laboratory by means of luminescence or ESR measurements, based on a comparison with a corresponding amount of artificially administered (usually mono-energetic $\beta$- or $\gamma$-) dose, which is called the equivalent dose. ESR and luminescence ages are derived from this palaeodose and the total environmental dose rate. The dose rate describes the location-specific strength of natural ionizing radiation per time and is formally defined as the rate at which energy is absorbed by a dosimeter from the flux of radiation to which the dosimeter is exposed (e.g., Aitken, 1998).

While the cosmic component of the environmental dose rate is typically derived from information on the exact sampling position by applying well established formulas (e.g., Prescott and Hutton, 1994), the contribution of ionizing radiation arising from the surrounding sediments is calculated by determining the activity concentrations of the relevant natural radionuclides. For dose rate determination, several in situ procedures using either portable gamma-spectrometers or sensitive dosimeters such as BeO or Al$_2$O$_3$ have been developed. Additionally, laboratory analyses of bulk material are applied, inter alia including emission counting methods such as thick source alpha counting (TSAC; e.g., Turner et al., 1958) and beta counting (e.g., Sanderson, 1988), spectrometric approaches like low-level high-resolution gamma spectrometry (HRGS) and neutron activation analysis (NAA) as well as geochemical techniques such as inductively coupled plasma mass spectrometry (ICP-MS) and inductively coupled plasma optical emission spectrometry (ICP-OES).

Recently, a laboratory-based, combined $\alpha$- and $\beta$-particle detection instrument called $\mu$Dose-system has been developed (e.g., Miłosz et al., 2017; Tudyka et al., 2018). Providing a cost-efficient approach, this novel device allows the determination of radionuclide concentrations of $^{238}$U, $^{235}$U, $^{232}$Th and $^{40}$K. Up to now, this measurement system has not been tested systematically. Therefore, we present a performance test based on three $\mu$Dose-devices and compare the results obtained with the

new approach with those from established analytical techniques. The comprehensive study includes measurements on a total of ~50 samples, covering natural samples as well as IAEA standards, and involves five different laboratories. In addition, we provide recommendations for sample handling and data analysis for the $\mu$Dose-results derived from practical experiences so far made in the Giessen Luminescence Laboratory.

## 2  The $\mu$Dose-system

### 2.1  Technical description

The $\mu$Dose-system (Fig. 1) is a compact and easy to handle analytical instrument allowing the simultaneous detection of $\alpha$- and $\beta$-particles. For this purpose, the system is equipped with a dual-layer scintillator (Fig. 1b) consisting of a plate of $\beta$-sensitive (synthetic) material, which is coated with a thin film of $ZnS : Ag$ for detecting $\alpha$-particles. This dual-layer scintillator is part of the cover plate of the sample container and thus placed between the sample material and the photomultiplier. Since the scintillator does not have direct contact to the sample material under investigation and is additionally protected by an approximately $0.2 \ \mu m$ thin silver foil, the scintillator is reusable. In addition, this silver foil reflects photons emitted by the scintillators which increases photon counting efficiency and guarantees an equal level of efficiency independent of the respective sample material. For measurements, the sample material is placed on a thin disc of filter paper, which is stored in a gas tight sample container (Fig. 1e). The diameter of the disc matches the diameter of the photomultiplier tube (PMT), which may vary from $30 \ mm$ to $70 \ mm$. For the present study, a PMT with a diameter of $70 \ mm$ was used. A detailed description of the technical setup is given by Tudyka et al. (2018).

$\alpha$- and $\beta$-particles are discriminated based on the different shapes of the pulses induced by the particles. Amplified by the PMT, these pulses are identified and analysed by a pulse analyser unit that has previously been described in detail by Miłosz et al. (2017). During the measurement process, an Analogue to Digital Converter (ADC) samples and transforms the incoming pulses into digital values (ADC values). These ADC values are time-stamped and stored in a database. Thus, a re-evaluation of data is possible at any time without the need to repeat the measurement. Data analyses is performed by applying a special algorithm. This algorithm determines pulse height and pulse shape of the stored pulses allowing the discrimination between $\alpha$- and $\beta$-induced pulses as well as the elimination of background pulses caused by interfering variables. Data analysis is possible after finishing the measurement as well as during a still running measurement process.

The $\mu$Dose-system is not only capable of discriminating between $\alpha$- and $\beta$-particles, but also allows the detection of decay pairs. Such decay pairs arise from the fast succession of two decays and thus two incoming pulses (pairs) detected within a very short and specific period of time. These pairs are the results of short half-lives of some members of the involved decay chains. This principle has long been used in TSAC to derive the particular contributions from the uranium and thorium series (e.g., Aitken, 1985). Whereas the TSAC technique is restricted to $\alpha$-$\alpha$-pairs, the $\mu$Dose-system is also able to make use of $\beta$-$\alpha$-pairs, which can be identified based on the individual timestamp of each detected pulse. Thus, the determination of the activities arising from the $^{238}$U-, $^{235}$U- and $^{232}$Th-series as well as from the decay of $^{40}$K is based on two $\alpha$-$\alpha$-pairs and two $\beta$-$\alpha$-pairs. A summary is given in Table 1. One $\alpha$-$\alpha$-pair is part of the $^{235}$U-series and caused by the successive $\alpha$-decays of

$^{219}$Rn and $^{215}$Po, with the latter showing a half-life of 1.78 ms. With $^{220}$Rn/$^{216}$Po (half-life of $^{216}$Po: 145 ms) a second $\alpha$-$\alpha$-pair is part of the $^{232}$Th-series. One $\beta$-$\alpha$-pair arises from the successive decay of $^{212}$Bi and $^{212}$Po, which has a half-life of only 299 ns. Finally, the $\beta$-decay of $^{214}$Po (half-life: 164 µs) following an $\alpha$-decay of $^{214}$Bi is a characteristic component of the $^{238}$U-series. On condition that the investigated sample is in or at least close to secular equilibrium, the $\alpha$- and $\beta$-counts associated with the above mentioned decay pairs allow to calculate the concentrations of $^{238}$U, $^{235}$U as well as $^{232}$Th and thus provide the possibility to derive the series-specific activities. The particular $^{40}$K activity is determined as residual value derived from the excess of observed $\beta$-counts over the $\beta$-counts expected to arise from the determined $^{238}$U-, $^{235}$U- and $^{232}$Th-series. For details on how decay pairs are statistically identified and for a thorough description of formulas and assumptions used for calculating the specific contributions arising from the different decay series, the reader is referred to Tudyka et al. (2018).

**Table 1.** Decay pairs used to derive the specific contributions arising from the $^{238}$U-, $^{235}$U- and $^{232}$Th-series as well as from $^{40}$K.

| Pair type | Radionuclides | Half-life | Series |
|---|---|---|---|
| $\alpha$-$\alpha$ | $^{219}$Rn/$^{215}$Po | 1.78 ms | $^{235}$U-series |
| $\alpha$-$\alpha$ | $^{220}$Rn/$^{216}$Po | 145 ms | $^{232}$Th-series |
| $\beta$-$\alpha$ | $^{212}$Bi/$^{212}$Po | 299 ns | $^{232}$Th-series |
| $\beta$-$\alpha$ | $^{214}$Bi/$^{214}$Po | 164 $\mu$s | $^{238}$U-series |

## 2.2 System calibration

Since the activities are derived from the net count rates of the detected decay pairs using equations for which pair-specific calibration parameters are needed (cf., Tudyka et al., 2018), these parameters have to be determined for each $\mu$Dose-device by performing calibration measurements on material of known activities. The $\mu$Dose-systems can be calibrated for different amounts of sample material using calibration material distributed by the manufacturer.

For the calibration of the $\mu$Dose-systems at the Giessen Luminescence Laboratory, three standards prepared on behalf of the IAEA are used, i.e. IAEA-RGU-1, IAEA-RGTh-1 and IAEA-RGK-1 (hereafter always mentioned as RGU-1, RGTh-1 and RGK-1). For a detailed description of the calibration material, see Sect. 3.1 of this article. Moreover, a device- and location-specific background value has to be determined using a background disc placed on the sample holder. Since all three calibration materials have high activities, the respective calibration measurements were performed for only 24 hours. For the background determination, a longer lasting measurement of seven days was executed. In order to increase the accuracy of the calibration, we advise to use repeated measurements of all standards and to derive the calibration parameters from the means of these repeated measurements. This will substantially reduce the impact of random errors potentially affecting single measurements. In the Giessen Luminescence Laboratory, the means of three repeated measurements for each standard and one background measurement are combined to define the device-specific calibration. Comprising 10 separate measurements (3x3 IAEA standards + 1 background measurement), the whole calibration procedure requires a total duration of ~14 days. The $\mu$Dose software offers a user-friendly calibration module to define and manage calibrations.

Since raw data of finished measurements (i.e. the ADC coded pulses) are stored in a database, data evaluation can be performed at any time using different calibration settings. This allows recalculating the determined activities without the need of conducting another time-consuming measurement. Furthermore, this database solution provides the opportunity to identify significant changes in the technical specifications of the devices.

Although there were no such significant changes detected so far during the ~1.5 years of $\mu$Dose-usage in the Giessen Luminescence Laboratory, such changes seem possible and might predominantly be attributed to various ageing effects. These ageing effects may affect the used silver foil, the dual-layer scintillator or other electronic components of the devices, in particular the efficiency of the built-in PMTs. Thus, we strongly recommend a re-calibration of the $\mu$Dose-systems at regular intervals in order to guarantee that the determined calibration parameters still match the actual technical status of the measurement setup. In the Giessen Luminescence Laboratory, a re-calibration of the $\mu$Dose-systems is performed twice a year with time intervals of not more than six to eight months. This re-calibration is not only based on an isolated measurement of a specific test sample, but comprises the whole calibration procedure as described above, including nine separate measurements of IAEA standards as well as a prolonged measurement of the device-specific background signal.

## 2.3 Determination of uncertainties

The $\mu$Dose-system considers several sources of uncertainties that are associated either with the measurement procedure or with the sample preparation. The most dominant uncertainties are derived from the counting statistics of calibration measurements (here IAEA standards and background) and investigated samples. Additionally, there is a relative counting rate uncertainty of 0.001 that corresponds to sample preparation reproducibility or other unknown sources of error. This component of uncertainty will not decrease with increasing measurement time. The $\mu$Dose-system allows adjusting the (recommended) default values for each device by user-specified values. Uncertainty propagation considers correlations between the individual uncertainties determined for the different radionuclide activities/concentrations. A detailed description on the mode of uncertainty propagation used for $\mu$Dose-analysis is provided by Tudyka et al. (2020).

## 3 Sample materials for the performance test

### 3.1 IAEA standards

Provided by the IAEA, RGU-1, RGTh-1 and RGK-1 standards were not only used as calibration material for the $\mu$Dose-systems (see above) but also for performance tests validating the quality of calibration. The RGU-1 and RGTh-1 standards were both prepared by the Canada Centre for Mineral and Energy Technology. The standards were derived from a uranium ore (BL-5) and a thorium ore (OKA-2), respectively. These raw materials were diluted with floated silica powder of negligible uranium and thorium contents. For both raw materials, the IAEA was able to show them to be in radioactive equilibrium (for details see IAEA, 1987).

The IAEA certifies the radionuclide concentrations as follows: $400 \pm 2 \, \mathrm{mg \cdot kg^{-1}}$ uranium, $< 1 \, \mathrm{mg \cdot kg^{-1}}$ thorium and $< 0.002 \, \%$ potassium for the RGU-1 standard; $6.3 \pm 0.4 \, \mathrm{mg \cdot kg^{-1}}$ uranium, $800 \pm 16 \, \mathrm{mg \cdot kg^{-1}}$ thorium and $0.02 \pm 0.01 \, \%$ potassium for the RGTh-1 standard (IAEA, 1987). For RGU-1, these concentrations correspond to radioactivity values of

145 $4{,}941 \pm 99 \, \mathrm{Bq \cdot kg^{-1}}$ for $^{238}$U, $224 \pm 5 \, \mathrm{Bq \cdot kg^{-1}}$ for $^{235}$U and negligible values for $^{232}$Th as well as for $^{40}$K. For RGTh-1, the IAEA gives values of $3{,}250 \, \mathrm{Bq \cdot kg^{-1}}$ (95 % C.I.: $3{,}160 - 3{,}340 \, \mathrm{Bq \cdot kg^{-1}}$) for $^{232}$Th, $3.6 \, \mathrm{Bq \cdot kg^{-1}}$ (95 % C.I.: $3.3 - 3.9 \, \mathrm{Bq \cdot kg^{-1}}$) for $^{235}$U, $78 \, \mathrm{Bq \cdot kg^{-1}}$ (95 % C.I.: $72 - 84 \, \mathrm{Bq \cdot kg^{-1}}$) for $^{238}$U and $6.3 \, \mathrm{Bq \cdot kg^{-1}}$ (95 % C.I.: $3.1 - 9.5 \, \mathrm{Bq \cdot kg^{-1}}$) for $^{40}$K (see datasheet on the IAEA homepage). All values are summarized in Table 2 (concentrations) and Table 3 (activities).

The RGK-1 standard was derived from high purity potassium sulphate supplied and certified under the label Extra Pure DAC by the Merck Company. Based on repeated measurements applying atomic absorption spectrometry, the potassium content was determined by the IAEA Laboratories Seibersdorf, which also estimated values for the uranium and thorium content (for details see IAEA, 1987). The RGK-1 standard reveals a $^{40}$K-activity of $14{,}000 \, \mathrm{Bq \cdot kg^{-1}}$ (95 % C.I.: $13{,}600 - 14{,}400 \, \mathrm{Bq \cdot kg^{-1}}$), showing negligible concentrations of thorium ($< 0.01 \, \mathrm{mg \cdot kg^{-1}}$) and uranium ($< 0.001 \, \mathrm{mg \cdot kg^{-1}}$).

**Table 2.** Radionuclide concentrations as certified by the IAEA (IAEA, 1987). Uranium and thorium values are given in $\mathrm{mg \cdot kg^{-1}}$, potassium is given in %. Uncertainties represent the 95 % C.I.

| Reference Material | Uranium [mg · kg⁻¹] | Thorium [mg · kg⁻¹] | Potassium [%] |
|---|---|---|---|
| RGK-1 | $< 0.001$ | $< 0.01$ | $44.8 \pm 0.3$ |
| RGTh-1 | $6.3 \pm 0.4$ | $800 \pm 16$ | $0.02 \pm 0.01$ |
| RGU-1 | $400 \pm 2$ | $< 1$ | $< 0.002$ |

**Table 3.** Recommended radionuclide-specific activities as provided on the homepage of the IAEA. All values are given in $\mathrm{Bq \cdot kg^{-1}}$. Please note that only those values were considered for this table for which information are provided by the IAEA. Uncertainties represent the 95 % C.I.

| Reference Material | $^{238}$U [Bq · kg⁻¹] | $^{235}$U [Bq · kg⁻¹] | $^{232}$Th [Bq · kg⁻¹] | $^{40}$K [Bq · kg⁻¹] |
|---|---|---|---|---|
| RGK-1 | NA | NA | NA | $14{,}000 \pm 400$ |
| RGTh-1 | $78 \pm 6$ | $3.6 \pm 0.3$ | $3{,}250 \pm 90$ | $6.3 \pm 3.2$ |
| RGU-1 | $4{,}941 \pm 99$ | $224 \pm 5$ | NA | NA |

## 3.2 Nussy loess standard

The Nussy reference material is a loess sample from a well-known loess section near Nußloch (e.g., Antoine et al., 2001; Bente and Löscher, 1987; Sabelberg and Löscher, 1978) located ~10 km south of the city of Heidelberg, at the eastern shoulder of the Upper Rhine Graben, Germany (49° 19' N, 8° 43' E, 217 m a.s.l.). Here, loess sediments revealing a total thickness of ~16 m are covering a basement of Middle Triassic limestone and dolomite formations. The sample was collected from the Upper Weichselian loess deposits accumulated during the last glacial-interglacial cycle. The Nussy sample reveals grain sizes characteristic for loess sediments, ranging from $2-63\,\mu m$. The material was first used as a reference material in the Heidelberg Luminescence Laboratory (e.g., Kalchgruber, 2002; Rieser, 1991) and prepared as the first certified reference material (CRM) for loess by Kasper et al. (2001). Based on an inter-laboratory comparison with contributions from three different laboratories, Preusser and Kasper (2001) provided the following concentrations, which were derived from the average of 11 HRGS measurements: $2.68 \pm 0.06\,\mathrm{mg\cdot kg^{-1}}$ (SD: $0.09\,\mathrm{mg\cdot kg^{-1}}$) for the total U content, $7.41 \pm 0.23\,\mathrm{mg\cdot kg^{-1}}$ (SD: $0.34\,\mathrm{mg\cdot kg^{-1}}$) for Th and $0.96 \pm 0.01\,\%$ (SD: $0.02\,\%$) for K. Later, these values were re-evaluated: Based on geochemical analyses involving over 50 laboratories, the International Association of Geoanalysts (IAG) characterized the Nussy loess as reference material IAG UoK Loess, reporting radionuclide concentrations of $2.80 \pm 0.20\,\mathrm{mg\cdot kg^{-1}}$ ($2\sigma$) for U and $8.12 \pm 0.25\,\mathrm{mg\cdot kg^{-1}}$ ($2\sigma$) for Th (for details see IAG, 2017). The IAG did not determine the potassium content. Most recently, Murray et al. (2018) re-investigated the Nussy loess standard applying HRGS. They reported activities of $37 \pm 2\,\mathrm{Bq\cdot kg^{-1}}$ for $^{238}U$, $35.5 \pm 0.5\,\mathrm{Bq\cdot kg^{-1}}$ for $^{232}Th$ and $369 \pm 5\,\mathrm{Bq\cdot kg^{-1}}$ for $^{40}K$. For our study, we refer to the values published by Preusser and Kasper (2001). All values are summarized in Table 4.

## 3.3 Volkegem loess standard

The Volkegem reference material is a loess sample that has been collected in a former quarry in the city of Volkegem (East-Flanders, Belgium). Originally, the reference material was characterized in a comprehensive study by De Corte et al. (2007). After drying at 110°C and milling, the sample material was sieved to grain diameters $< 50\,\mu m$ and homogenized. This material was investigated applying $k_0$-INAA and HPGe gamma spectrometry and additionally cross-checked by in situ gamma spectrometry, TSAC and Geiger-Muller beta-counting (for a detailed description see De Corte et al., 2007). As reference data, they were able to determine mean radionuclide concentrations of $2.79 \pm 0.12\,\mathrm{mg\cdot kg^{-1}}$ for U, $10.4 \pm 0.6\,\mathrm{mg\cdot kg^{-1}}$ for Th and $1.65 \pm 0.15\,\%$ for K as well as mean activities of $36.1 \pm 1.7\,\mathrm{Bq\cdot kg^{-1}}$ for $^{235+238}U$, $42.2 \pm 2.5\,\mathrm{Bq\cdot kg^{-1}}$ for $^{232}Th$ and $497 \pm 45\,\mathrm{Bq\cdot kg^{-1}}$ for $^{40}K$. Like the Nussy reference material, Murray et al. (2018) also re-investigated the Volkegem standard. They report slightly higher activities of $37.8 \pm 0.7\,\mathrm{Bq\cdot kg^{-1}}$ for $^{238}U$, $44.2 \pm 0.5\,\mathrm{Bq\cdot kg^{-1}}$ for $^{232}Th$ and $570 \pm 5\,\mathrm{Bq\cdot kg^{-1}}$ for $^{40}K$. For our study, we refer to the original values published by De Corte et al. (2007). All values are summarized in Table 4.

**Table 4.** Summary of concentrations and activities published for the Nussy and Volkegem loess standards. Values used as reference values for this study are highlighted as bold numbers. The errors for these values represent 95 % C.I.s. Please note: For better comparison, the standard deviations (SD) given by Preusser and Kasper (2001) were translated to 95 % C.I. values in this table and in the text, applying a t-distribution with 10 degrees of freedom. Concentration and activity values not provided by the original publications were re-calculated for this table and are flagged by **-symbols (for details see table notes). Simulated environmental dose rates based on assumed constant values for water content and cosmic radiation were derived from the radionuclide concentrations in this table (for details see table notes).

| | Concentrations | | | Activities | | | | Simulated environmental dose rates* |
|---|---|---|---|---|---|---|---|---|
| | Uranium [mg·kg⁻¹] | Thorium [mg·kg⁻¹] | Potassium [%] | $^{238}$U [Bq·kg⁻¹] | $^{235}$U [Bq·kg⁻¹] | $^{232}$Th [Bq·kg⁻¹] | $^{40}$K [Bq·kg⁻¹] | [Gy·ka⁻¹] |
| *Nussy loess standard* | | | | | | | | |
| Preusser and Kasper (2001) | **2.68 ± 0.06** | **7.41 ± 0.23** | **0.96 ± 0.01** | 33.1 ± 0.74** | 1.54 ± 0.03** | 30.1 ± 0.93** | 304 ± 3.2** | **1.93 ± 0.07** |
| Murray et al. (2018) | 3.00 ± 0.16** | 8.75 ± 0.12** | 1.17 ± 0.02** | 37 ± 2 | 1.72 ± 0.09** | 35.5 ± 0.5 | 369 ± 5 | 2.25 ± 0.09 |
| IAG (2017) | 2.80 ± 0.20 | 8.12 ± 0.25 | NA | 34.6 ± 2.50** | 1.60 ± 0.10** | 32.9 ± 1.00** | NA | NA |
| *Volkegem loess standard* | | | | | | | | |
| De Corte et al. (2007) | **2.79 ± 0.12** | **10.4 ± 0.6** | **1.65 ± 0.15** | 34.5 ± 1.5 | 1.59 ± 0.09 | 42.2 ± 2.5 | 497 ± 45 | **2.71 ± 0.15** |
| Murray et al. (2018) | 3.06 ± 0.06** | 10.9 ± 0.12** | 1.80 ± 0.02** | 37.8 ± 0.7 | 1.76 ± 0.03** | 44.2 ± 0.5 | 570 ± 5 | 2.92 ± 0.11 |

* Values in this column represent simulated environmental dose rates calculated for the $90 - 200$ μm grain size fraction of HF-etched quartz using the Dose Rate and Age Calculator DRAC v1.2 (Durcan et al., 2015). All calculations are based on the radionuclide concentrations provided in this table, applying the dose rate conversion factors given by Guérin et al. (2011). Please note that a constant water content of $15 \pm 5$ % and a constant contribution of $0.150 \pm 0.015$ Gy $\cdot$ ka$^{-1}$ arising from cosmic radiation were assumed for all calculations. We would like to point out that these assumed values do not correspond to the values that might actually be determined for Nussy and Volkegem sampling sites. Therefore, the calculated dose rates are referred to as 'simulated environmental dose rates' in the table and in the text.

** Values converted from radionuclide concentrations to activities and vice versa were calculated using conversion factors and natural uranium composition given by Guérin et al. (2011). All calculations were verified applying the radionuclide-specific conversion factors provided in Table 2.5 of IAEA (2003).

 ## 3.4 Natural samples

For this study, 47 natural samples covering a great variety of environmental settings and landscapes were analysed in order to validate the performance of $\mu$Dose-measurements. The samples were provided by and measured in five laboratories in Germany and Poland, including the luminescence laboratories at the universities of Bayreuth, Cologne, Giessen and Heidelberg as well as the Institute of Physics in Gliwice. All analysed samples are summarized in Table B1 in Appendix B. A detailed description of sampling locations including geological, stratigraphic and morphological settings is provided in Appendix C.

## 4 Experimental settings for the $\mu$Dose-measurements in the Giessen Luminescence Laboratory

### 4.1 Sample preparation

All analysed samples were dried in a drying chamber at an elevated temperature of 105°C for several days. The dried sample material was gently crushed using a porcelain mortar and then homogenized. Approximately 10 g of this homogenized material were pulverized in a ball mill (Retsch M 400) using a frequency of 29.5 Hz for 45 minutes and dry sieved with analytical sieves showing mesh sizes of 63 µm. This sieving procedure is used as an additional backstop in the sample preparation, which is based on the idea that coarse-grained residuals of $> 63$ µm indicate that the applied milling duration was not sufficient to provide fully pulverized material. Thus, the sieving step is not used to exclude resilient grains with diameters $> 63$ µm, since this would cause a mineral-specific fractionation and introduce bias to the $\mu$Dose-measurements. The additional sieving step merely aims at surveying the quality of the preparation procedure applied in the Giessen Luminescence Laboratory. With respect to the samples investigated in this study, we were not able to detect any residual material $> 63$ µm. Therefore, we conclude that the applied milling duration of 45 minutes was sufficient to provide pulverized material adequate for $\mu$Dose-measurements.

After weighing 3.00 g of this pulverized material with a high precision balance (Fig. 1c), the sample material was placed on a sample carrier and carefully fixed on top of a disc of filter paper, using a stamp made of acrylic glass (Fig. 1a & Fig. 1d). The discs show diameters matching the diameters of the used PMTs (here 70 mm). For the measurement procedure, the filled sample carriers were stored in a device-specific, gas-tight measurement container (Fig. 1e) which prevents migration of radon from and into the container.

Additionally, the bottom of the measurement container is filled with granular active carbon, which contributes to reducing the radon concentration of the air within the container. This aims at avoiding an accumulation of radon gas right in front of the scintillator module, which may impact the alpha count rate.

### 4.2 Technical settings for the $\mu$Dose-devices

All measurements have been performed on $\mu$Dose-devices installed in the Luminescence Laboratory of the Department of Geography at the Justus Liebig University Giessen. The devices are situated in a laboratory that is exclusively designated for sedimentological analyses and for the preparation of dose rate samples. Thus, neither luminescence measurement systems with

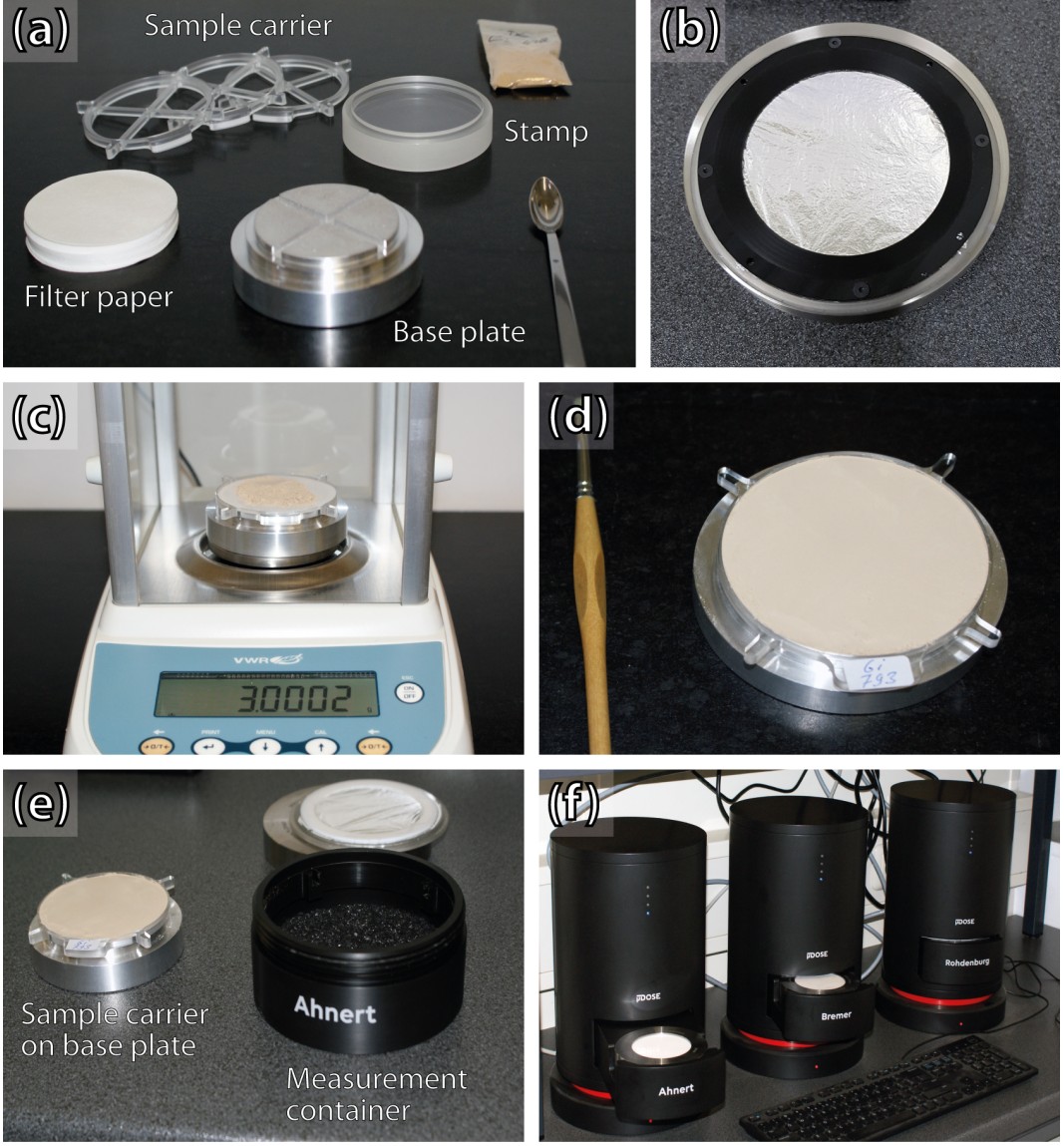

**Figure 1.** Photos showing the $\mu$Dose-devices and equipment: (a) Sample carrier and equipment for sample preparation. (b) Scintillator unit with silver foil. (c) High-precision balance used for weighing 3.00 g of sample material. (d) Prepared sample on a sample carrier with a diameter of 70 mm. (e) Prepared sample material and measurement container. (f) Three $\mu$Dose-devices installed in the Giessen Luminescence Laboratory.

their integrated radioactive sources nor other technical devices that might generate radiation fields or electromagnetic fields had any kind of potentially distracting impact on the $\mu$Dose-measurements.

Three measurement systems with identical technical features are installed – named "005-Ahnert", "006-Bremer" and "007-Rohdenburg" (Fig. 1f). All devices are equipped with internal high voltage power supplies and photomultiplier tubes (PMT) that have a photocathode diameter of 70 mm. The measurement units are controlled by a single PC with distinct, system-specific measurement software. Measurement data are primarily stored on the built-in SSD-drive of this PC and additionally saved on backup servers provided by the Department of Geography. A device-specific unique measurement ID is assigned for each measurement.

The $\mu$Dose-systems at the Giessen laboratory are calibrated for a total amount of 3 g of sample material. In order to guarantee that all investigated samples matched this specification, the samples were checked using a high precision balance prior to the measurements. Only those samples lying within a range of 2.995 g to 3.005 g were accepted for measurement.

In order to minimize the possible bias of $\alpha$-counts due to the adhesion of radon bearing particles from ambient air, a delayed start of the measurement procedure is advised. In the Giessen Luminescence Laboratory, the applied time delay was at least one hour, i.e. after storing the sample in the measurement container and sealing it, the operator has to wait for at least one hour before initiating the start of the measurement procedure. For ease of use, upcoming versions of the $\mu$Dose software will provide the possibility to define an automated and user specified time delay.

The respective measurement times strongly depended on the sample-specific activities. For the experiment analysing the impact of measurement duration (see Sect. 4.3.2), various measurement times were applied. Due to their high activities, relatively short measurement times of ~24 hours were used for the IAEA-standards RGU-1, RGTh-1 and RGK-1, yielding excellent counting statistics. For the remaining samples, including both loess standards and natural samples, measurements were continued until the number of detected $\alpha$-counts reached the level of approximately 3,000 counts, an empirically determined threshold that was derived from long lasting experiences with TSAC at the University of Bayreuth (pers. comm. L. Zöller). Depending on the respective activities of a sample, this value corresponds to measurement durations of two to four days for samples revealing average environmental dose rates in the range of $2\ \mathrm{Gy} \cdot \mathrm{ka}^{-1}$ to $4\ \mathrm{Gy} \cdot \mathrm{ka}^{-1}$.

## 4.3 Experimental setups

For this study, a total of three different experiments were conducted which aimed at assessing the performance and reliability of the $\mu$Dose-systems.

### 4.3.1 Accuracy and reproducibility of results

A first experimental setting aimed at assessing the reproducibility and accuracy of measurement results gained with the $\mu$Dose-systems. Therefore, repeated measurements were performed on the certified IAEA standards and on the two loess standards. For these measurements, one 3 g subsample of each standard was prepared. These subsamples were used for all measurements on all devices. So, there was no re-sampling. Once stored in the device-specific measurement container, the subsamples were not removed from the container until all measurements on the respective device were completed. Measurements for the IAEA standards were restricted to ~24 hours, while the loess standards were each measured for approximately four to five days. Measurements have been performed on all three devices.

### 4.3.2 The impact of the measurement duration

The measurement duration required for a reliable result might be a crucial point since accuracy and precision of the $\mu$Dose-measurements strongly depend on the net count rates of $\alpha$- and $\beta$-particles. In TSAC, device-specific numbers of $\alpha$-counts are often used as thresholds to ensure count rates that enable the calculation of radionuclide concentrations with a sufficiently high precision. As already mentioned above, a value of approximately $3,000$ $\alpha$-counts is routinely used in the Giessen laboratory to guarantee reliable results. However, this value is merely an arbitrary threshold, which is derived from long lasting experiences with TSAC in the luminescence laboratory at the University of Bayreuth (pers. comm. Ludwig Zöller). With particular respect to environmental samples revealing low radionuclide concentrations the usage of such a high threshold may lead to prolonged measurement times that would not be desirable for routine dose rate measurements. In the Giessen Luminescence Laboratory for instance, several samples originating from the Negev desert (Israel) were measured, for which dose rates of $< 1\ \mathrm{Gy} \cdot \mathrm{ka}^{-1}$ could be determined. Applying the $3,000$ $\alpha$-counts criterion, each sample had to be measured for more than 15 days.

In order to investigate the impact of measurement duration and to test whether shorter measurement times also provide reliable results, 3 g subsamples of both loess standards Nussy and Volkegem were repeatedly measured applying various measurement times. The measurement times lasted from a minimum of approximately ten hours to more than seven days, corresponding to total $\alpha$-counts of ~200 to more than $8,000$. All measurements were performed as stand-alone measurements, i.e. the results for short- and medium-time measurements were calculated from numerous separate measurements and not derived from one long-lasting master-measurement. Both subsamples were measured on all three $\mu$Dose-systems. Once stored in the measurement container, the subsamples were not removed from the container until all measurements were finished for the respective device. For all measurements, the same subsamples of Nussy and Volkegem loess standards were used.

### 4.3.3 Comparison to established measurement procedures

In order to test the overall performance of the $\mu$Dose-system, we initiated a comprehensive inter-laboratory comparison including five different laboratories from Germany and Poland, which applied different measurement procedures. The involved laboratories were: (i) the Giessen Luminescence Laboratory, (ii) the Bayreuth Luminescence Laboratory, (iii) the Cologne Luminescence Laboratory, (iv) the Heidelberg Luminescence Laboratory and (v) the Institute of Physics (Division of Geochronology and Environmental Isotopes) in Gliwice.

For this performance test, we re-investigated a total of 47 environmental samples for which either radionuclide concentrations or activities had already been determined by either TSAC in combination with ICP-OES (Bayreuth) or low-level HRGS (Cologne, Heidelberg, Gliwice). Details on sample preparation and technical specifications of the $\mu$Dose-systems in Giessen are provided in Sect. 4.1 and 4.2. The measurement configurations applied in the other participating laboratories are briefly summarized in Table 5. Details of sample preparation and information on the applied measurement procedures including used gamma lines are provided in Appendix A. The investigated samples represent a broad variety of regions and environmental settings (see Table B1 and sample characterization in the Appendix).

**Table 5.** Summary of measurement settings in the participating laboratories. Details of sample preparation and applied measurement procedures are described in the text and in Appendix A.

| | Bayreuth Luminescence Laboratory | Cologne Luminescence Laboratory | Heidelberg Luminescence Laboratory | Gliwice Institute of Physics |
|---|---|---|---|---|
| Method(s) | TSAC / ICP-OES | Low-level HRGS | Low-level HRGS | Low-level HRGS |
| Device(s) | Littlemore Low Level Alpha Counter 7286<br><br>Varian Vista-Pro$^{TM}$ | Ortec Coaxial Profile M7080-S GEM HPGe Detector<br><br>Canberra Coaxial Profile GC4040 Ge Detector | Broad Energy Ge Detector Canberra BE 2020 | Extended Range Coaxial Ge Detector Canberra GX 4518 |
| Drying procedure | Several days at 105°C | ≥ 2 days at 50°C | Several days at 50°C | Several days depending on water content |
| Amount of sample | ~5 g | 200 g / 590 g | 30 g | 100 g |
| Storage time | ≥ 4 weeks | ≥ 4 weeks | ≥ 4 weeks | ≥ 3 weeks |
| Measurement duration | ≥ 3,000 $\alpha$-counts | ≥ 42 hours | - | ≥ 24 hours |
| Calibration and quality control | Tony loess standard | Nussy loess standard and artificially irradiated samples | Nussy loess standard | RGU-1, RGTh-1, RGK-1, IAEA-385 |

## 5 Results and discussion

### 5.1 Accuracy and reproducibility of measurement results

The accuracy and reproducibility of measurement results were tested by repeated measurements of three certified IAEA standards that had also been used for the calibrations of the $\mu$Dose-devices. Due to their high radionuclide concentrations these standards provide high decay rates improving the statistics of $\alpha$- and $\beta$-counts. Figure 2 shows the results of repeated measurements of these standards expressed as relative deviations of measured results from the expected reference values provided by the IAEA. For the plot, only the results obtained for the dominant radioactive emitter of the respective standard were con-

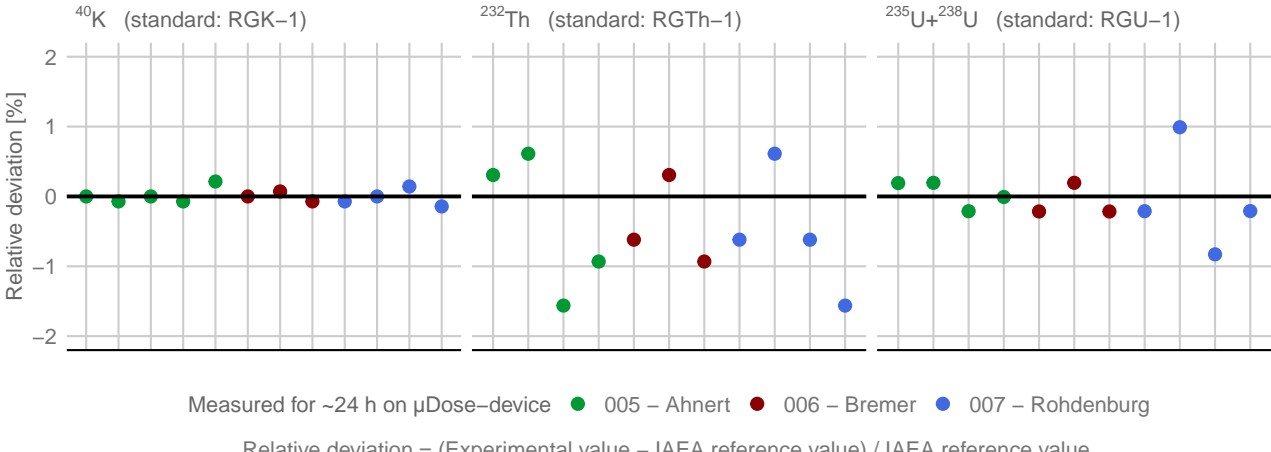

**Figure 2.** Results for repeated measurements of the investigated IAEA standards. The different colours of the symbols represent three different measurement devices (see legend). All plots show the relative deviation of measured values from the respective reference values provided for the IAEA standards. Sample RGK-1 is illustrated on the left, the thorium standard RGTh-1 is shown in the centre and RGU-1 is depicted on the right. Bold lines illustrate the 0 %-deviation (i.e., a perfect agreement of measured and expected values). Please note that only activities arising from the dominant radioactive emitter of the respective standard were considered for this figure.

sidered. So, for RGK-1 only the activity of $^{40}$K, for RGTh-1 the activity of $^{232}$Th and for the uranium standard RGU-1 the combined activities of $^{235}$U and $^{238}$U were analysed.

From the results shown in Fig. 2 we are able to draw two important conclusions: (i) $\mu$Dose-measurements of IAEA standards reveal an excellent accuracy. For potassium, thorium and uranium, all measured values are within the respective 95 % confidence intervals certified by the IAEA. The majority of relative deviations of measured activities from the certified values are < 1 %. The mean relative calibration deviations are: $-0.0001$ % for $^{40}$K, $-0.4554$ % for $^{232}$Th and $-0.0298$ % for $^{235+238}$U. These values correspond to measured-to-given ratios of 1.0000 for $^{40}$K, 0.9955 for $^{232}$Th and 0.9997 for $^{235+238}$U and indicate an excellent quality of the implemented $\mu$Dose calibrations. (ii) The repeated measurements of IAEA standards are characterized by an excellent reproducibility. The determined results reveal neither statistically significant outliers nor distinct differences between the different measurement devices. The relative standard deviations (RSD) obtained from statistics and averaged for all devices are: 0.10 % for $^{40}$K, 0.80 % for $^{232}$Th and 0.45 % for $^{235+238}$U. An overview summarizing accuracy and statistical reproducibility is provided in Table 6.

These results may be attributed to the high content of radionuclides characteristic for the investigated IAEA standards. Although only measured for ~24 hours, the net $\alpha$-counts detected for RGU-1 and RGTh-1 show mean values of ~46,000 cts and ~30,000 cts, respectively. These total numbers of $\alpha$-counts are more than 10 times higher than the threshold value of

**Table 6.** Accuracy and precision of $\mu$Dose-measurements of certified IAEA standards. The accuracy is expressed as measured-to-given ratios (MGR). Precision is given as relative standard deviation (RSD) of measured acitivities. Only results derived for the dominant emitter of the respective IAEA standard were considered for this table.

| Radionuclide (IAEA Standard) | Measured-to-given ratios (MGR) and relative standard deviations (RSD) for... | | | |
| --- | --- | --- | --- | --- |
| | ...all devices | ...device Ahnert | ...device Bremer | ...device Rohdenburg |
| $^{40}$K (RGK-1) | MGR: 1.0000 RSD: 0.10 % (N = 12) | MGR: 1.0001 RSD: 0.12 % (N = 5) | MGR: 1.0000 RSD: 0.07 % (N = 3) | MGR: 0.9998 RSD: 0.12 % (N = 4) |
| $^{232}$Th (RGTh-1) | MGR: 0.9955 RSD: 0.80 % (N = 11) | MGR: 0.9962 RSD: 1.02 % (N = 4) | MGR: 0.9959 RSD: 0.64 % (N = 3) | MGR: 0.9946 RSD: 0.89 % (N = 4) |
| $^{235+238}$U (RGU-1) | MGR: 0.9997 RSD: 0.45 % (N = 11) | MGR: 1.0004 RSD: 0.19 % (N = 4) | MGR: 0.9992 RSD: 0.24 % (N = 3) | MGR: 0.9994 RSD: 0.76 % (N = 4) |

3,000 $\alpha$-counts typically applied in the Giessen Luminescence Laboratory for $\mu$Dose-measurements of sediment samples. In summary, these results indicate the excellent quality of $\mu$Dose-calibration and a good reproducibility of measurements.

Figure 3 shows the accuracy and reproducibility of results gained for the two loess standards Nussy (Preusser and Kasper, 2001) and Volkegem (De Corte et al., 2007). For the Nussy standard, the mean values of the determined concentrations averaged over all three devices are: 1.08 % (SD: 0.07 %) for potassium, 8.53 mg $\cdot$ kg$^{-1}$ (SD: 1.30 mg $\cdot$ kg$^{-1}$) for thorium and 2.43 mg $\cdot$ kg$^{-1}$ (SD: 0.32 mg $\cdot$ kg$^{-1}$) for uranium. These values correspond to mean measured-to-given-ratios of 1.13 for potassium, 1.15 for thorium and 0.91 for uranium.

For the Volkegem loess standard, the averaged values of all $\mu$Dose-measurements are as follows: 1.66 % (SD: 0.03 %) for potassium, 12.25 mg $\cdot$ kg$^{-1}$ (SD: 1.53 mg $\cdot$ kg$^{-1}$) for thorium and 2.53 mg $\cdot$ kg$^{-1}$ (SD: 0.32 mg $\cdot$ kg$^{-1}$) for uranium. The corresponding measured-to-given-ratios are: 1.00 for potassium, 1.18 for thorium and 0.91 for uranium.

For both samples, the uranium contents are slightly underestimated by ~10 %, whereas thorium contents are overestimated by ~15 % and ~18 %, respectively. For the Nussy standard, potassium is also overestimated by ~13 % while there is a nearly perfect agreement with the reference value for the Volkegem standard.

At a first glance, the results obtained for the loess standards seem to indicate some kind of problems concerning the accuracy of the $\mu$Dose-measurements. In order to check this and to assess intra-sample variability, we re-sampled and re-measured both loess standards. The results of these additional measurements did not significantly differ from the results reported in this study and showed similar deviations of ~9 % up to ~17 %. However, when talking about deviations determined for specific radionuclides, it should be considered that uranium and thorium concentrations are not detected independently in $\mu$Dose-measurements (see Sec. 2.1). This dependency can clearly be seen when looking at the Th- and U-concentrations of

**Figure 3.** Results from repeated $\mu$Dose-measurements for the loess standards Nussy (upper part) and Volkegem (lower part). The different colours of the symbols represent three different measurement devices (see legend). All plots show radionuclide concentrations either in $\mathrm{mg \cdot kg^{-1}}$ (U and Th) or in % (K). Please note that the bold reference lines indicate radionuclide contents originally published for the Nussy loess standard by Preusser and Kasper (2001) and for the Volkegem loess standard by De Corte et al. (2007). Dashed lines characterize the corresponding 95 % C.I. Error bars indicate measurement uncertainties on the $2\sigma$-level.

the Volkegem loess standard in the lower part of Fig. 3. Whenever Th-concentrations are higher than the expected value, the corresponding U-concentration is lower and vice versa. For the Nussy loess standard, the results shown in the upper part of
Fig. 3 are similar, but not as obvious as for the Volkegem loess standard. When deriving environmental dose rates, the exact Th/U-ratio has some relevance. However, the conversion of alpha count rates to dose rates in TSAC shows that the conversion factor for the beta dose rate is higher for uranium and lower for thorium, while the conversion factor for the gamma contribution is higher for thorium and lower for uranium (e.g., Aitken, 1985). In the end, there is at least a partial compensation. As a result, the total environmental dose rate does not vary much with the exact Th/U-ratio (e.g., Li and Tso, 1995). With respect to the
determination of environmental dose rates, deviations in the individual concentrations/activities of uranium and thorium are acceptable as long as the combined activity arising from uranium and thorium is close to the expected value.

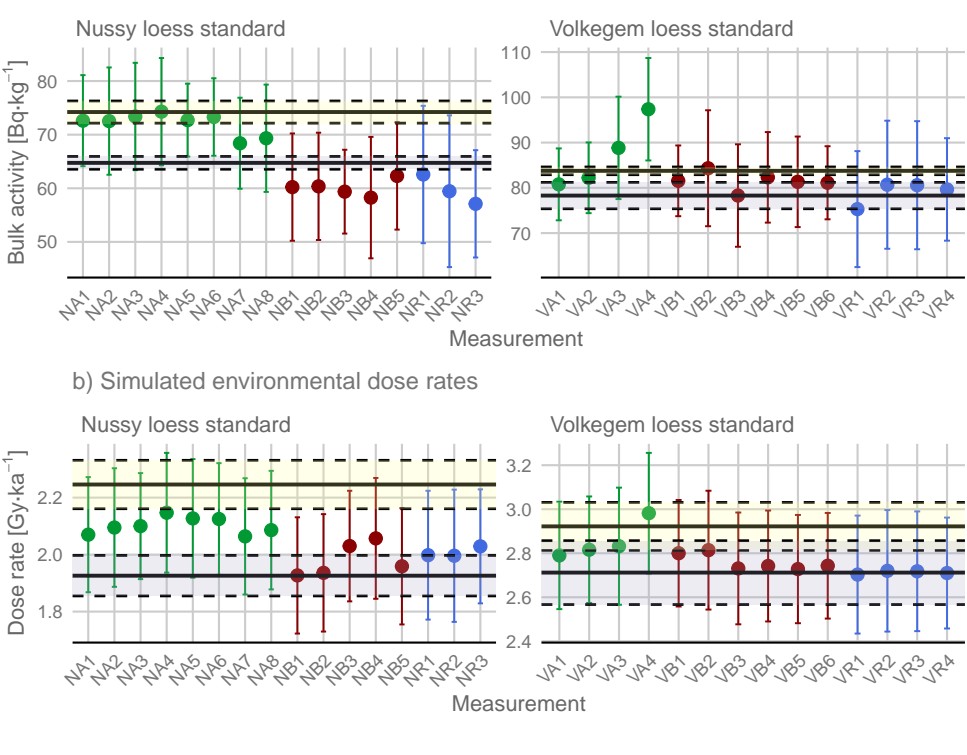

**Figure 4.** Bulk uranium and thorium activities given in $\mathrm{Bq \cdot kg^{-1}}$ (a) and simulated environmental dose rates given in $\mathrm{Gy \cdot ka^{-1}}$ (b) for both investigated loess standards. The Nussy loess standard is depicted on the left, the Volkegem loess standard is shown on the right. The different colours of the symbols represent three different measurement devices (see legend). Please note that the bold reference lines centred within the grey area indicate radionuclide contents originally published for the Nussy loess standard by Preusser and Kasper (2001) and for the Volkegem loess standard by De Corte et al. (2007) whereas the bold lines centred within the yellow area represent benchmark values derived from the results published by Murray et al. (2018).

Figure 4a shows the combined activity arising from the uranium and thorium decay chains for the Nussy loess standard (left) and for the Volkegem loess standard (right). With respect to the latter, the values determined with the $\mu$Dose-system are in good agreement with the expected benchmark value published by De Corte et al. (2007). With individual measured-to-given ratios ranging from $0.96$ (measurement VR1) to $1.24$ (measurement VA4), the mean measured-to-given ratio averaged for all devices is $1.05$. Revealing a relative standard deviation of ~6 %, the spread in data is rather low, although the combined uranium and thorium activities determined for measurement VA3 and VA4 show rather large deviations. Based on the 3 IQR-criterion these values can be characterized as extreme outliers. Not considering these values, the measured-to-given ratio for the Volkegem loess standard would average out at $1.03$.

This overall good results are reflected by the calculated simulated environmental dose rates for the Volkegem loess standard, which is depicted on the right side of Fig. 4b. Based on the radionuclide concentrations published by De Corte et al. (2007), a value of $2.71 \pm 0.15$ Gy $\cdot$ ka$^{-1}$ is expected. The simulated environmental dose rates calculated for the $\mu$Dose-results show a range from 2.70 Gy $\cdot$ ka$^{-1}$ to 2.98 Gy $\cdot$ ka$^{-1}$ and average at a value of $2.77 \pm 0.02$ Gy $\cdot$ ka$^{-1}$, which corresponds to a mean measured-to-given ratio of 1.02. If the above mentioned extreme outliers are not considered for data analysis, the average

simulated dose rate for the remaining measurements is $2.75 \pm 0.01$ Gy $\cdot$ ka$^{-1}$ and the measured-to-given ratio improves to 1.01. In summary, we can conclude that the $\mu$Dose-measurements are able to provide results that allow the calculation of simulated environmental dose rates that are in good agreement with the expected benchmark value for the Volkegem loess standard.

      For the Nussy loess standard the results are less satisfying. With an average combined uranium and thorium activity of

66.01 Bq $\cdot$ kg$^{-1}$, the $\mu$Dose-measurements overestimate the benchmark of 64.74 Bq $\cdot$ kg$^{-1}$ derived from the values published by Preusser and Kasper (2001) by only ~2 %. This would correspond to a promising overall measured-to-given ratio of 1.02. However, the bulk uranium and thorium values determined by the $\mu$Dose-measurements show a rather large relative standard deviation of ~10 %. Furthermore, there are distinct inter-device differences reflected by pronounced variations in the device-specific mean measured-to-given ratios. These ratios range from 0.92 for the devices "006-Bremer" and "007-Rohdenburg" to

1.13 for device "005-Ahnert". While the first two devices underestimate the expected value, the latter shows a considerable overestimation.

      When looking at the calculated simulated environmental dose rates, the results are slightly better than for the combined activities of uranium and thorium. The mean value averaged for all measurements is $2.04 \pm 0.02$ Gy $\cdot$ ka$^{-1}$ and is slightly higher than the benchmark of $1.93 \pm 0.07$ Gy $\cdot$ ka$^{-1}$. With device-specific measured-to-given ratios of 1.09 (Ahnert), 1.03 (Bremer) and 1.04 (Rohdenburg), the average measured-to-given ratio for all devices corresponds to 1.06. Except for the

values of two $\mu$Dose-measurements, all simulated environmental dose rates are beyond the range of the 95 % confidence interval given for the benchmark of Preusser and Kasper (2001). But still, all simulated environmental dose rates are within the range of benchmarks calculated for the IAG values and for the values provided by Murray et al. (2018), which can clearly be seen on the left side of Fig. 4b.

For a meaningful interpretation of results it has to be considered that the published reference values were derived from a limited number of gamma spectrometry and k$_0$-INAA measurements that were carried out under specific laboratory conditions. Therefore, they may suffer from distinct methodological problems. On a closer inspection, it thus becomes apparent that inter-methodological deviations of more than 10 % are neither unusual for dosimetry measurements (e.g., Murray et al., 2015) nor necessarily indicate serious deficits in the respective measurement procedures. On contrary, the results obtained for the IAEA

standards (see above) suggest good accuracy and reproducibility of $\mu$Dose-measurements.

      A closer look at the publication of Preusser and Kasper (2001) shows that the authors do not only provide results derived from HRGS, but also ICP-MS based values from three different laboratories. The magnitude of scatter in the data reported for the Nussy loess standard is comparable to the maximum deviations determined for the $\mu$Dose-measurements. For the K content, values from 0.96 % to a maximum of 1.14 % are reported, while the Th and U contents range from 7.4 mg $\cdot$ kg$^{-1}$

to $8.8 \, \text{mg} \cdot \text{kg}^{-1}$ and from $2.3 \, \text{mg} \cdot \text{kg}^{-1}$ to $2.7 \, \text{mg} \cdot \text{kg}^{-1}$, respectively. Referring to the reference value for the Nussy loess standard, this spread in data corresponds to relative deviations of approximately 15 % to 19 %.

A smaller, but still considerable spread in the determined data can be observed, when the values published by Preusser and Kasper (2001) are compared to the IAG reference values for U and Th. Here, the IAG values exceed the originally published data by ~5 % (U) and ~10 % (Th).

A similar finding can be noticed for Volkegem loess activities given by De Corte et al. (2007) when compared to results derived from the re-measurements of Murray et al. (2018). For all radionuclides, Murray et al. (2018) reported substantially higher activities. While the $^{232}$Th-activity exceeds the originally determined value by ~5 %, the deviations for $^{238}$U and $^{40}$K are considerably more pronounced revealing relative values of ~10 % and ~15 %, respectively.

**Table 7.** Results from $\mu$-Dose-measurements and reference values for K, Th and U contents of Nussy loess standard (upper part) and Volkegem loess standard (lower part). The values for K are given in %, the values for U and Th are given in $\text{mg} \cdot \text{kg}^{-1}$. Reference values (and their associated 95 % C.I.s) are according to Preusser and Kasper (2001) and De Corte et al. (2007). The 95 % C.I.s for Nussy have been recalculated based on the SD-values provided by Preusser and Kasper (2001). Uncertainties of the $\mu$Dose-measurements correspond to 95 % C.I.s. The table shows mean values for individual $\mu$Dose-devices as well as average values calculated as mean of all measurements on the three devices.

| Radionuclide | Reference value | Average value for all devices | Mean contents Ahnert | Mean contents Bremer | Mean contents Rohdenburg |
|---|---|---|---|---|---|
| *Nussy loess standard* | | | | | |
| K [%] | $0.96 \pm 0.01$ | $1.08 \pm 0.04$ | $1.04 \pm 0.02$ | $1.11 \pm 0.11$ | $1.15 \pm 0.15$ |
| Th [$\text{mg} \cdot \text{kg}^{-1}$] | $7.41 \pm 0.23$ | $8.53 \pm 0.69$ | $9.51 \pm 0.76$ | $7.86 \pm 0.75$ | $7.03 \pm 2.08$ |
| U [$\text{mg} \cdot \text{kg}^{-1}$] | $2.68 \pm 0.06$ | $2.43 \pm 0.17$ | $2.60 \pm 0.22$ | $2.18 \pm 0.23$ | $2.39 \pm 1.01$ |
| | | (N = 16) | (N = 8) | (N = 5) | (N = 3) |
| | | | | | |
| *Volkegem loess standard* | | | | | |
| K [%] | $1.65 \pm 0.15$ | $1.66 \pm 0.02$ | $1.67 \pm 0.07$ | $1.66 \pm 0.03$ | $1.64 \pm 0.05$ |
| Th [$\text{mg} \cdot \text{kg}^{-1}$] | $10.4 \pm 0.6$ | $12.25 \pm 0.88$ | $13.28 \pm 3.32$ | $12.63 \pm 0.42$ | $10.65 \pm 1.05$ |
| U [$\text{mg} \cdot \text{kg}^{-1}$] | $2.79 \pm 0.12$ | $2.53 \pm 0.19$ | $2.59 \pm 0.69$ | $2.33 \pm 0.02$ | $2.76 \pm 0.49$ |
| | | (N = 14) | (N = 4) | (N = 6) | (N = 4) |

## 5.2 Measurement time and associated alpha count rates

Dosimetry measurements can be time-consuming. This might either be caused by the need of extensive preparation procedures and long-lasting storage times or due to the measurement process itself. For the $\mu$Dose-system, sample preparation is relatively rapid and samples can be measured immediately after the preparation procedure without the need for storage for specific periods of time. Since accuracy and precision of $\mu$Dose-measurements strongly depend on the net alpha and beta count rates,

the measurement duration is a decisive factor for the quality of the obtained results. In terms of net $\alpha$- and $\beta$-counts, this becomes obvious when comparing the results gained from the investigated IAEA standards (up to ~$30,000 - 46,000$ $\alpha$-counts) to the results determined for the loess standards (up to ~$3,000$ $\alpha$-counts; see Sect. 5.1). In theory, longer measurement times will provide better counting statistics (i.e., higher numbers of $\alpha$- and $\beta$-counts) which should improve both, accuracy and precision of the results. From a theoretical point of view, long lasting measurements thus should be favoured. However, it is obviously impossible to implement such an approach in practice since for typical environmental samples trying to reach count rates similar to those reported for the IAEA standards would mean having to accept long lasting measurements of several weeks or even months.

Figure 5 shows the results of an experiment aiming at identifying whether there is a particular lower limit of measurement durations for which still reliable results can be expected. The plots show radionuclide concentrations (y-axis) plotted against the total number of detected $\alpha$-counts (x-axis). All measurements were conducted as separate stand-alone measurements on the same subsamples of the Nussy and Volkegem loess standards.

The majority of results is clustering rather closely to the median values indicated by the bold lines. Overall, this seems to be true for all measurement durations. For the thorium and uranium contents of the Volkegem loess standard, short-time measurements with a total number of $\alpha$-counts $< 2,000$ show a larger deviation from the median. This also applies to extremely short measurements of only few hours for U- and Th-values obtained for the Nussy standard. Apart from that, other short-time measurements for Nussy do not show such a distinct deviation from the median, but only reveal a slightly larger scatter compared to long-time measurements. With respect to the potassium results, the picture is not so clear. For Volkegem, short-time measurements of $< 2,000$ $\alpha$-counts at least show a large scatter and a slightly larger deviation from the median than measurements with longer durations. For Nussy however, neither the deviation from the median nor the inter-measurement scatter indicate that this group of measurements might be less precise than measurements of longer duration. Unlike for thorium and uranium, even measurements with a duration of only some hours do not differ from the median value.

Although there are some sources of uncertainty which do not get smaller with time (see Sect. 2.3), longer lasting measurements in theory should be expected to be associated with considerably smaller uncertainties due to better counting statistics. In summary, our results are confirming this relationship, which might be derived from Fig. 5 and becomes quite obvious when looking at the average measurement uncertainties for different groups of measurements arranged by their respective durations (expressed by their total number of $\alpha$-counts), which are summarized in Table 8.

Overall, the measurement uncertainties are reduced by longer measurement times. This applies to both loess standards and to all radionuclides. The biggest reduction, however, is observed when comparing short-time measurements of $< 2,000$ $\alpha$-counts to those showing a total number of $\alpha$-counts of $2,000 - 4,000$ (i.e. medium-time measurements). For the Nussy loess standard for instance, relative reductions of uncertainties of ~8 % (K), ~44 % (Th) and ~45 % (U) are achieved. With 4 % (K), 29 % (Th) and 29 % (U) similar but smaller relative reductions in uncertainties can be determined for the Volkegem loess standard when short-time and medium-time measurements are compared.

A further increase to long measurement durations corresponding to more than $4,000$ $\alpha$-counts (long-time measurements) further reduces the uncertainties, yet typically not to the same extent as for the medium-time measurements. For the Nussy

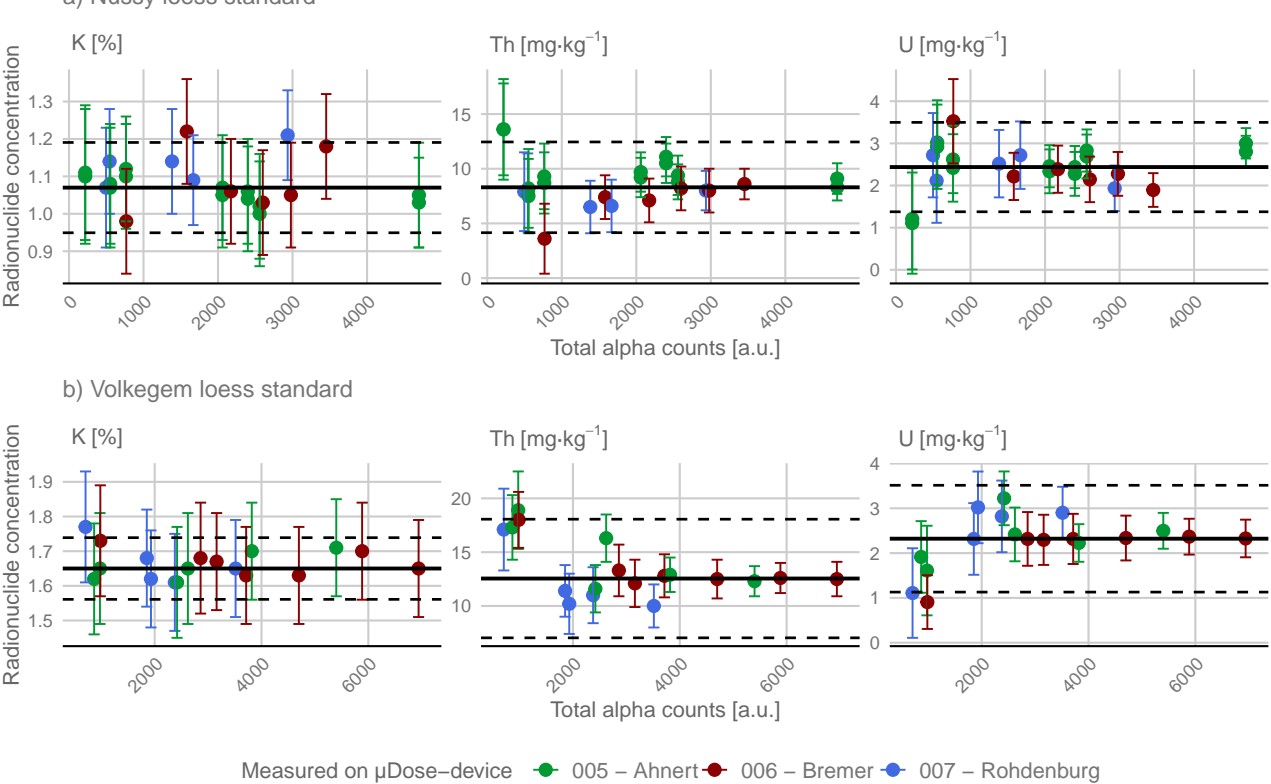

**Figure 5.** Results from $\mu$Dose-measurements of two loess standards (Nussy = upper part; Volkegem = lower part). Values on the x-axis represent the number of total $\alpha$-counts. Y-axis-values give the respective radionuclide concentration either in % (K) or in $\mathrm{mg} \cdot \mathrm{kg}^{-1}$ (Th and U). The different colours of the symbols represent three different $\mu$Dose-devices (see legend). The bold lines illustrate the median values derived from the determined results. Dashed lines indicate the $2\sigma$-deviation.

loess standard, prolonged measurements of $> 4{,}000$ $\alpha$-counts correspond to relative reductions of the original (short-time)

uncertainties of 13 % (K), 59 % (Th) and 60 % (U). Particularly for U and Th, these values are only slightly higher than those of the reduction for medium-time measurements. With total relative reductions of 9 % (K), 49 % (Th) and 48 % (U) compared to the short-time measurements, similar results can be found for the Volkegem loess standard.

In Fig. 6 the obtained results for radionuclide concentrations are illustrated as box-whisker-plots. This allows identifying statistically relevant outliers which were determined based on the 1.5 interquartile-range- (IQR-)criterion, i.e. the difference of

430 the third and the first quartile of the whole data set as shown by the box, extended both in the lower and upper direction by a factor 1.5 * IQR as illustrated by the whiskers. Values outside this range are highlighted by red circles and labelled with their respective measurement durations expressed as the total number of $\alpha$-counts. From Fig. 6 it can be concluded that the majority of outliers arises from short-time measurements of $< 2{,}000$ $\alpha$-counts, which equals measurement durations of approximately

**Table 8.** Averaged uncertainties for $\mu$Dose-measurements of loess standards Nussy (upper part) and Volkegem (lower part) grouped by their respective total numbers of $\alpha$-counts.

| Radionuclide | Duration < 2,000 $\alpha$-counts | Duration 2,000 - 4,000 $\alpha$-counts | Duration > 4,000 $\alpha$-counts |
|---|---|---|---|
| *Nussy loess standard* | | | |
| K [%] | 0.075 | 0.069 | 0.065 |
| Th [mg $\cdot$ kg$^{-1}$] | 1.617 | 0.900 | 0.650 |
| U [mg $\cdot$ kg$^{-1}$] | 0.465 | 0.255 | 0.190 |
| *Volkegem loess standard* | | | |
| K [%] | 0.077 | 0.074 | 0.070 |
| Th [mg $\cdot$ kg$^{-1}$] | 1.517 | 1.075 | 0.775 |
| U [mg $\cdot$ kg$^{-1}$] | 0.417 | 0.295 | 0.215 |

one day or only a few hours. Only three medium-time measurements revealing $\alpha$-counts of ~2,400, ~2,900 and ~3,500 have been identified as outliers. Therefore, we conclude that the probability of obtaining results not consistent with the average values is higher for short-time measurements showing a total number of $\alpha$-counts of less than 2,000.

For Fig. 7 the data were grouped according to measurement durations, which illustrates the impact of measurement time even more evidently and supports the conclusions drawn from Fig. 5 and 6. With respect to the uranium and thorium contents of the Volkegem loess standard (Fig. 7 lower part), medium- and long-time measurements agree rather well. For uranium, the median values are $2.35$ mg $\cdot$ kg$^{-1}$ (long) and $2.37$ mg $\cdot$ kg$^{-1}$ (medium) with associated relative standard deviations (RSD) of $3$ % and $14$ %, respectively. For thorium, median values of $12.5$ mg $\cdot$ kg$^{-1}$ (RSD = $9$ %; long) and $12.5$ mg $\cdot$ kg$^{-1}$ (RSD = $15$ %, medium) were derived. These group medians are identical within errors and reveal rather small intra-group scatter (at least when compared to the short-time group). For the short-time measurements, the results are completely different. Here, median values of $1.76$ mg $\cdot$ kg$^{-1}$ (RSD = $43$ %) for uranium and $17.2$ mg $\cdot$ kg$^{-1}$ (RSD = $24$ %) for thorium were calculated. These median values differ clearly from those determined for either the long-time or the medium-time group. For uranium, the short-time measurements underestimate the medium- and long-time measurements by ~25 %. For thorium, an overestimation of ~38 % can be observed. With respect to the results obtained for potassium, the picture is not as clear as for uranium and thorium. The median values (short: $1.67$ mg $\cdot$ kg$^{-1}$; medium: $1.65$ mg $\cdot$ kg$^{-1}$; long: $1.68$ mg $\cdot$ kg$^{-1}$) show rather good agreement. Only the slightly larger scatter in data observed for the short-time measurements (RSD = $4$ %) compared to the medium- (RSD = $2$ %) and long-time (RSD = $2$ %) groups suggests that the short-time measurements might not provide reliable results (see also Table 9).

For the Nussy loess standard (Fig. 7, upper part), the results are more difficult to interpret. The median values indicate differences between the groups of measurement duration. However, the results summarized in Table 9 (upper part) are not

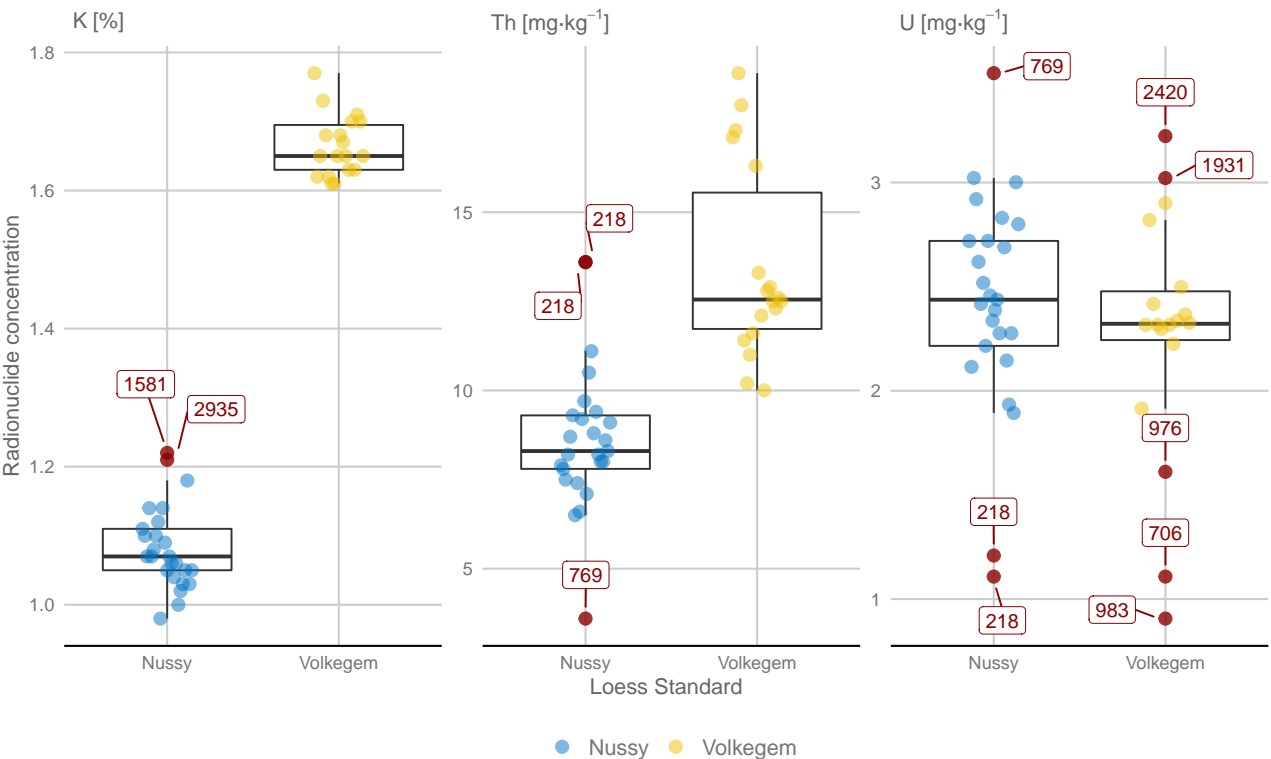

**Figure 6.** Radionuclide concentrations determined by $\mu$Dose-measurements given as % for K (left) and as $\mathrm{mg \cdot kg^{-1}}$ for Th (centre) and U (right). The different colours represent the different loess standards investigated (see legend). Outliers (red circles) were identified based on the 1.5 IQR-criterion and labelled with their respective number of total $\alpha$-counts.

as evident as for the Volkegem loess standard. Potassium contents calculated for long-time and medium-time measurements agree very well (long: 1.04 %; medium: 1.05 %), whereas the short-time value of 1.10 % is deviating from these two values. However, the relative deviation is only ~6 %. For thorium, we have a similar result. The median values of the medium- and long-time measurements are identical within errors, but do not significantly deviate from the results obtained for the short-time group which is slightly underestimating (~10 %) the results calculated for the other two groups. For uranium, the long-time measurements are slightly overestimating (~13 %) while short-time and medium-time groups show rather good agreement. With respect to the median values, the results are suggesting that the short-time measurements might be problematic. However, the evidence is not as clear as for the Volkegem loess standard. Showing values of 29 % and 34 % for uranium and thorium, respectively, at least the RSDs are rather large for the short-time measurements. Here, medium- and long-time groups show distinct lower RSDs of 12 % and 13 % (medium) as well as 4 % and 7 % (long). However, this does not apply to potassium for

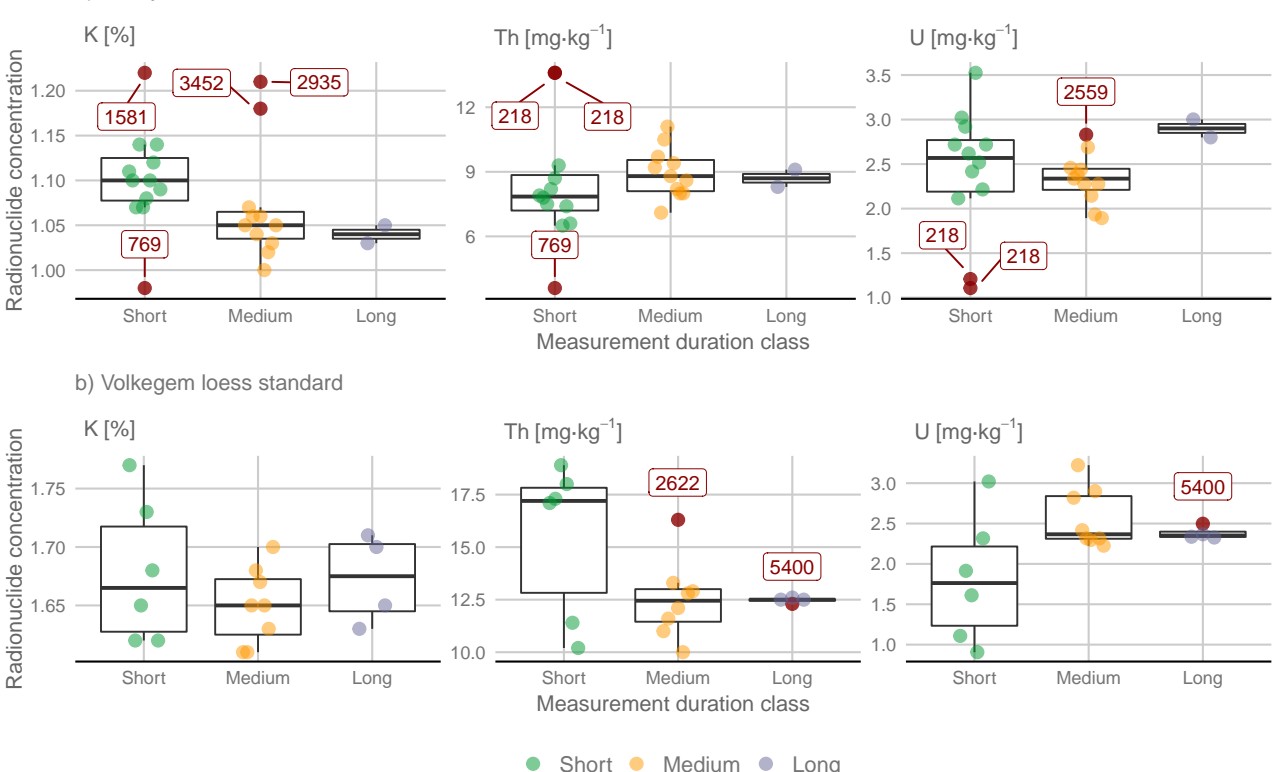

**Figure 7.** Radionuclide concentrations determined by $\mu$Dose-measurements given as $\%$ for K (left) and as $\mathrm{mg \cdot kg^{-1}}$ for Th (centre) and U (right). Results of individual measurements and boxplots for the loess standards Nussy (top) and Volkegem (bottom). Data grouped by measurement duration in three classes: short-time ($< 2{,}000$ $\alpha$-counts; green symbols); medium-time ($2{,}000 - 4{,}000$ $\alpha$-counts; yellow symbols); long-time ($> 4{,}000$ $\alpha$-counts; grey symbols). Outliers (red symbols) as identified by the 1.5 IQR-criterion and labelled with their respective numbers of total $\alpha$-counts. Classification not based on specific statistical arguments but reflecting the realisation of the experiments.

which a RSD of only $5\,\%$ could be determined for the short-time measurements. With respect to the outliers identified based on the 1.5 IQR-criterion, the majority belongs to short-time measurements of $< 2{,}000$ $\alpha$-counts.

Finally, there seems not to be a straightforward answer to the question whether there is a particular lower limit of measurement durations for which still reliable results can be expected. Our findings suggest that short-time measurements hold the greatest risk of providing results not in agreement with results obtained by longer-lasting measurements. This might be interpreted as an indicator of an unreliable measurement setup. At least, this is true for very short measurement durations of less than one day which should therefore be avoided. However, since our findings are somehow contradictory and might even point to a more or less sample-specific pattern, this conclusion should be regarded as a conservative rule of thumb.

**Table 9.** Median values and relative standard deviations (RSD) for radionuclide concentrations of the loess standards Nussy (upper part) and Volkegem (lower part) derived from $\mu$Dose-measurements. The individual measurements were classified in three groups of measurement durations based on the total number of $\alpha$-counts (short-time group: $< 2,000$ $\alpha$-counts; medium-time group: $2,000 - 4,000$ $\alpha$-counts; long-time group: $> 4,000$ $\alpha$-counts).

| Radionuclide | Short-time group | | Medium-time group | | Long-time group | |
| --- | --- | --- | --- | --- | --- | --- |
| | Median | RSD | Median | RSD | Median | RSD |
| *Nussy loess standard* | | | | | | |
| K | 1.10 % | 5 % | 1.05 % | 6 % | 1.04 % | 1 % |
| Th | 7.85 mg·kg$^{-1}$ | 34 % | 8.80 mg·kg$^{-1}$ | 13 % | 8.70 mg·kg$^{-1}$ | 7 % |
| U | 2.56 mg·kg$^{-1}$ | 29 % | 2.34 mg·kg$^{-1}$ | 12 % | 2.90 mg·kg$^{-1}$ | 4 % |
| *Volkegem loess standard* | | | | | | |
| K | 1.67 % | 4 % | 1.65 % | 2 % | 1.68 % | 2 % |
| Th | 17.2 mg·kg$^{-1}$ | 23 % | 12.5 mg·kg$^{-1}$ | 15 % | 12.5 mg·kg$^{-1}$ | 1 % |
| U | 1.76 mg·kg$^{-1}$ | 100 % | 2.37 mg·kg$^{-1}$ | 14 % | 2.35 mg·kg$^{-1}$ | 3 % |

In summary, we conclude that reliable results for the loess standards investigated in this study could be obtained by $\mu$Dose-measurements revealing total numbers of $\alpha$-counts of $2,000$ to $4,000$. For our samples this number of $\alpha$-counts corresponds to measurement durations of approximately two to four days (also see Table D1 in Appendix D). Extremely short measurement durations delivering $\alpha$-counts $< 2,000$ should be avoided due to insufficient counting statistics. Despite the benefit of further reducing measurement uncertainties, prolonged measurements of more than five days (i.e. $> 4,000$ $\alpha$-counts) are normally not necessary to ensure results of reasonable accuracy and precision. Since the counting statistic strongly depends on the sample-specific activity, we advise to use the total number of $\alpha$-counts as an indicator for an adequate measurement duration. In our experiments, samples (Nussy and Volkegem) measured for approximately two to four days revealed a mean number of ~$2,400$ $\alpha$-counts. Therefore, we suggest a threshold value of ~$2,500$ $\alpha$-counts as a minimum value in order to guarantee reliable measurement results.

### 5.3 $\mu$Dose-system performance for environmental samples

So far, the performance of the $\mu$Dose-system has only been tested on one synthetic sample with known activity composed as a mixture of different IAEA standards and on a very limited number of natural loess and archaeological samples (cf. Tudyka et al., 2018, 2020). In order to assess the performance of the $\mu$Dose-system for natural samples on a broader data basis, we carried out a series of inter-laboratory comparisons including TSAC, ICP-OES and low-level HRGS measurements. As our primary aim was to assess the potential of the $\mu$Dose-system to produce reliable data for calculating dose rates of samples with low radionuclide contents typical of natural environments, a total number of 47 samples from various environmental settings were re-measured on the $\mu$Dose-devices at the Giessen Luminescence Laboratory.

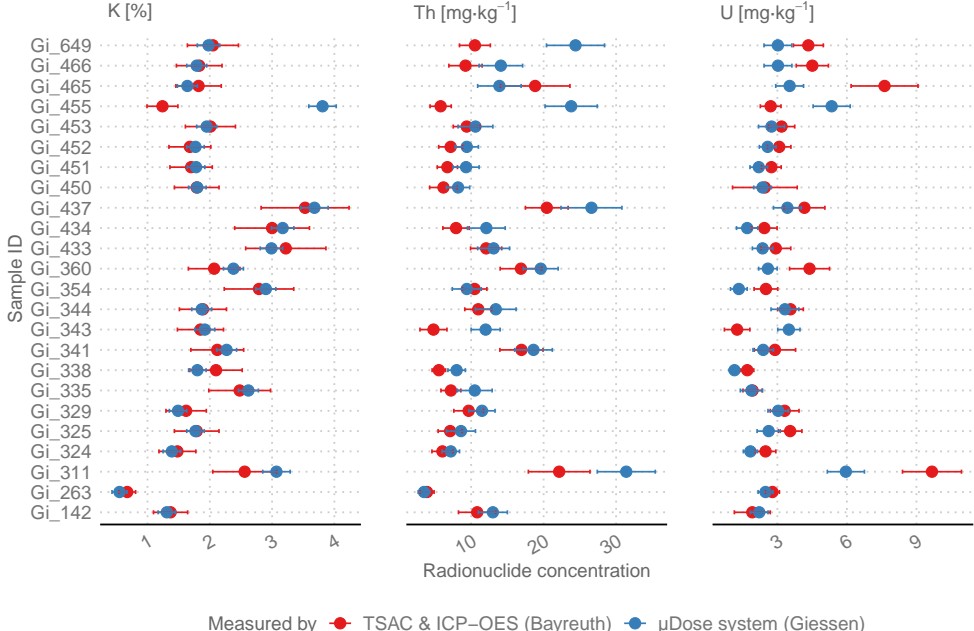

**Figure 8.** Comparison of results obtained by TSAC (U and Th) in combination with ICP-OES (K) (red symbols) to the findings derived from $\mu$Dose-measurements (blue symbols). Please note that the values are given as radionuclide concentrations ( % for K; $\mathrm{mg \cdot kg^{-1}}$ for U and Th).

Figure 8 shows the results for samples that were measured at the University of Bayreuth, applying TSAC for the determination of uranium and thorium contents and ICP-OES for potassium. For most samples, the findings indicate a very good agreement between the values derived from $\mu$Dose-measurements (blue symbols) and those obtained by TSAC and ICP-OES (red symbols). For uranium and thorium contents, the majority of samples agree within the $2\sigma$-level (U: 63 %, Th: 79 %). The calculated potassium contents often show a perfect match. 95 % of the investigated samples are within the $2\sigma$-level of agreement, 83 % even within the limits of $1\sigma$.

However, there are also some samples for which the determined values – particularly the determined contents of uranium and thorium – do not coincide on the $2\sigma$-level. Among these problematic samples are Gi311, Gi343, Gi360, Gi455, Gi465, Gi466 and Gi649. With respect to the last four of these samples, this pronounced difference of TSAC and $\mu$Dose values might be attributed to the possible presence of radioactive disequilibria caused by chemical and/or physical differentiation processes potentially affecting long-living members in the U and Th decay chains such as $^{234}$U, $^{230}$Th, $^{226}$Ra, $^{228}$Th and $^{228}$Ra (e.g., Degering and Degering, 2020; Krbetschek et al., 1994). This explanation is based on the specific context of the respective sampling locations. All four samples originate from Holocene fluvial flood plain sediments covering Pleistocene gravel beds.

For such sediments, strongly alternating ground water levels are characteristic. Generally, sediments exposed to fluctuating ground water levels are regarded as typical candidates for radioactive imbalances (e.g., Degering and Degering, 2020; Olley et al., 1996, 1997) since they are subjects of various translocation processes and potentially significant periodic changes in fundamental environmental conditions such as the pH value. With respect to the differing chemical properties of the individual elements in the decay chains, such imbalances can take several and complex forms, which may manifest either in a loss or in an accumulation of specific parent and daughter nuclides (e.g., Prescott and Hutton, 1995). Therefore, it appears not unlikely that the samples mentioned above suffer from distinct increases and/or decreases of particular radioactive daughter nuclides in the U and Th decay chains. Regardless of the specific nature of these potential imbalances, their existence would violate a central assumption of the specific algorithms used by the $\mu$Dose-system, which would most probably cause inadequate results for the calculated activities.

For the other samples, a lack of secular equilibrium might also be a suitable explanation for the detected deviations of measurement values. This might at least be true for samples Gi311 and Gi343. Both are colluvial samples which were taken from locations within profiles that were identified in the field as M-Go horizons according to the German soil classification system (Ad-Hoc-AG Boden, 2005). These horizons showed typical features of a gleysol revealing inter alia a characteristic accumulation of sesquioxides, which indicate periodical impact of ground water. As a result, secular disequilibria appear to be possible for these samples.

Figure 9 illustrates the results for the comparison of $\mu$Dose-measurements (blue symbols) with low-level HRGS (red symbols) performed in different laboratories. Figure 9a shows the results for the samples from the Heidelberg Luminescence Laboratory. On average, the obtained values are characterized by rather small discrepancies between $\mu$Dose-results and HRGS. The majority of Heidelberg samples agrees with the $\mu$Dose-results within either the $2\sigma$-level (U: 88 %; Th: 88 %; K: 88 %) or even within the $1\sigma$-level (U: 50 %; Th: 50 %; K: 75 %).

Figure 9b shows various samples that were measured at the Gliwice laboratory. Apart from samples provided by the Gliwice laboratory itself, these measurements also included some samples provided by the Giessen Luminescence Laboratory, which had previously been measured at the University of Bayreuth applying TSAC and ICP-OES. With respect to these latter samples, the results gained in Gliwice largely confirm the findings already discussed for the comparison of $\mu$Dose-measurements to TSAC and ICP-OES. For samples Gi311, Gi453 and Gi360, there is again a pronounced deviation of the $\mu$Dose-results to the independently obtained data. Sample Gi437, which was just within the limit of $2\sigma$-deviation for the TSAC-comparison, did not conform on the $2\sigma$-level when compared to the results from Gliwice. Particularly, this applies to the activities arising from $^{232}$Th and $^{238}$U. However, with respect to sample Gi455, the situation is different. While this sample showed the largest differences for the comparison to TSAC and ICP-OES, the values obtained by HRGS reveal a $2\sigma$-agreement with the $\mu$Dose-results. A straightforward interpretation of this finding is hardly possible, but it casts doubt on the above suggested explanation that Gi455 might suffer from a distinct radioactive disequilibrium. In fact, the extraordinary large discrepancies observed for Gi455 in the TSAC/ICP-OES comparison and the good agreement of $\mu$Dose-results and low-level HRGS values might rather indicate a serious problem during the TSAC/ICP-OES measurements. Particularly the amount of discrepancy observed for Gi455 is supporting this interpretation since other samples originating from the same sampling location (Gi450-Gi453) do

**Figure 9.** Comparison of results obtained by HRGS in different laboratories (red symbols) to the findings derived from $\mu$Dose-measurements (blue symbols). (a) Samples provided by the Heidelberg Luminescence Laboratory. Please note that these values are given as concentrations (% for K; $\mathrm{mg} \cdot \mathrm{kg}^{-1}$ for U and Th). For Gliwice and Cologne Laboratories values are given as activities ($\mathrm{Bq} \cdot \mathrm{kg}^{-1}$). Only the $^{238}$U-specific activity is shown for the samples measured in the Gliwice laboratory (b), while the combined activity of $^{235}$U and $^{238}$U is depicted for the samples from Cologne (c).

not show similar discrepancies. Furthermore, Gi455 was identified as a sample originating from floodplain loams of the Lahn river (see detailed description of sample materials in Appendix C). Based on long lasting experience with sediments from the Lahn catchment in the Giessen Luminescence Laboratory, floodplain material from the Lahn catchment is expected to show significant higher concentrations of thorium and uranium than material originating from fluvial gravels of the region. However, the TSAC/ICP-OES results obtained for Gi455 are in the same order of magnitude as the results obtained for Gi450-Gi453, which originate from the underlying terrace gravels. In the end we cannot be sure whether the distinct deviations observed for Gi455 were caused by problems during the TSAC/ICP-OES measurements or whether they can be explained by the presence of a radioactive disequilibrium.

Overall, the $2\sigma$-level proportions of agreement for all samples measured in Gliwice (including those from Giessen) are: 64 % (U), 50 % (Th) and 64 % (K). At a first glance, this could be misinterpreted as indication of serious methodological shortcomings. However, it has to be kept in mind that these measurements included a large number of samples from the Giessen laboratory which were previously identified as potentially problematic. Although the HRGS measurements in Gliwice did not give clear evidence of radioactive disequilibria, the presence of such disequilibria seems to be likely for at least 8 out of 14 measured samples when the specific sampling locations are considered.

Restricting the analysis to those five samples provided by the Institute of Physics in Gliwice for which no radioactive disequilibria were expected, the results are completely different. Except for sample U1_19, all samples reveal a very good or even excellent agreement with the $\mu$Dose-results from Giessen. On the $2\sigma$-level, the proportions of agreement between HRGS and $\mu$Dose are 80 % for K and Th and 100 % for U. So far, we were not able to find any reasonable explanation for the pronounced deviation of K and Th activities determined for sample U1_19.

With respect to the samples from the Cologne Luminescence Laboratory, the findings are also very good. Except for the potassium contents of three samples (COL_GGW1, COL_GGW6 and COL_UGW1) for which a distinct difference in the respective values is obvious, all values show excellent agreement with the $\mu$Dose-results. But also 50 % of the results for $^{40}$K conform on the $2\sigma$-level. For the activity of $^{235+238}$U, 90 % of the determined values agree on the $2\sigma$-level and still 60 % coincide within $1\sigma$. For $^{232}$Th, activities determined by $\mu$Dose and HRGS show a nearly perfect match. 100 % of the values agree within $2\sigma$ and still 70 % within $1\sigma$.

Surprisingly, this is also true for four samples for which radioactive disequilibria had been identified (COL_UGW1 – COL_UGW4). With respect to $^{235+238}$U and $^{232}$Th activities, a 100 % proportion of agreement on the $1\sigma$-level can be derived from the data, and for $^{40}$K still 50 %. In theory, the algorithm applied by the $\mu$Dose-software should not yield correct results since a major assumption of this algorithm is violated in the presence of radioactive disequilibria. As a consequence, we should expect large discrepancies between the applied methods since the determination of radionuclide activities in low-level HRGS and in the $\mu$Dose-system are based on differing approaches. Yet, our findings suggest that radioactive disequilibria are not necessarily associated with such large inter-methodological discrepancies. Although such discrepancies were detected for some of the analysed natural samples, this did obviously not apply to samples COL_UGW1 to COL_UGW4. A convincing explanation for this inconsistency can hardly be found at this moment. The findings for the Cologne samples are only based on a limited number of samples and are not supported by results obtained from the comparisons to the other laboratories (cf.

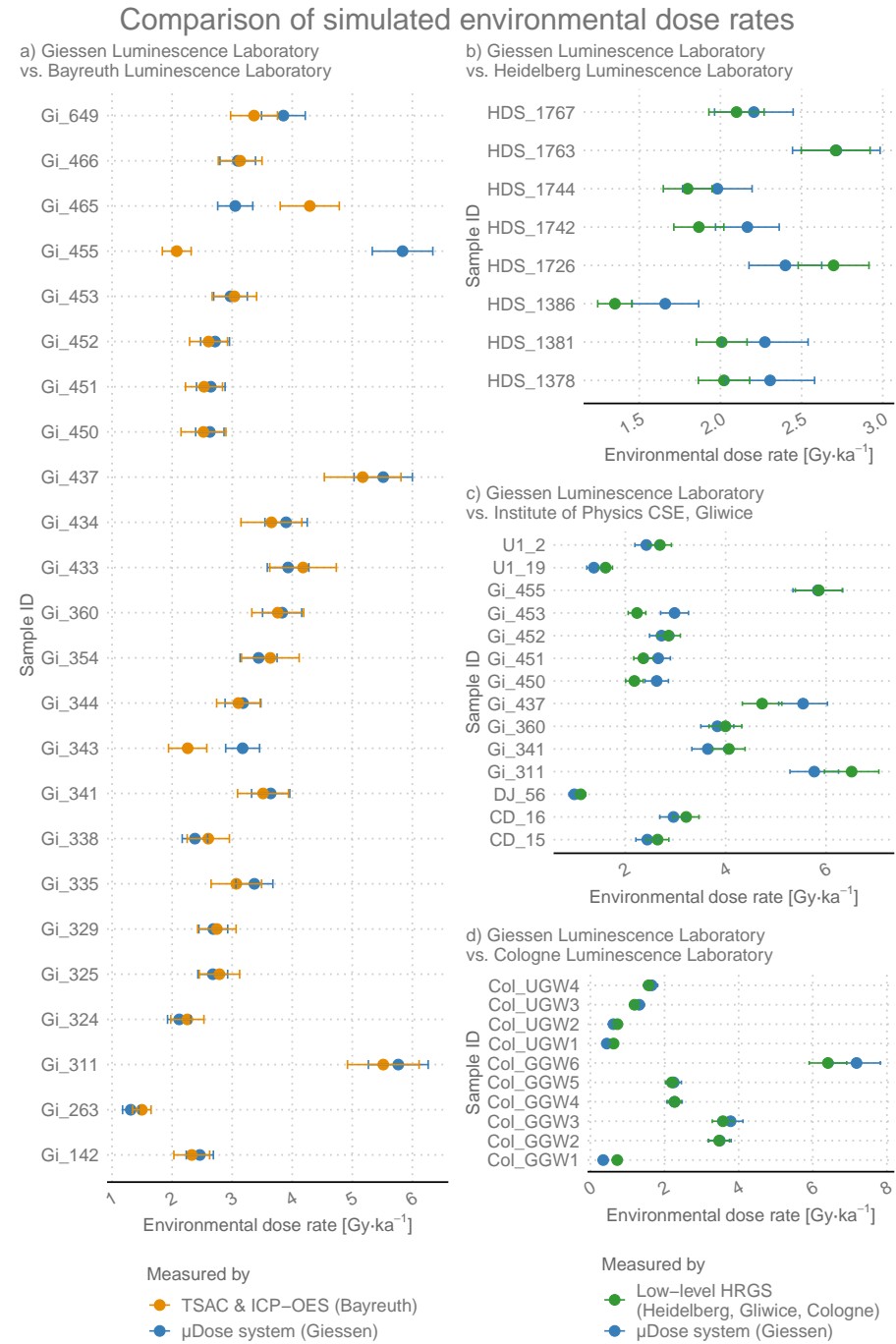

**Figure 10.** Comparison of simulated environmental dose rates for various natural samples. Assuming a constant water content of $15 \pm 5\,\%$ and a constant cosmic radiation of $0.150 \pm 0.015\,\mathrm{Gy \cdot ka^{-1}}$, all values were calculated for the $90 - 200\,\mu\mathrm{m}$ grain size fraction of HF-etched quartz using DRAC v1.2 (Durcan et al., 2015). Please be aware that these calculated values do not correspond to the actual dose rates and are thus referred to as 'simulated environmental dose rates'. For details the reader is referred to the table notes of Table 4.

Bayreuth and Gliwice). In the end, it can not be excluded that the results obtained for the four Cologne samples only match by chance. At the moment, we cannot decide whether these results are only odd anomalies or whether they represent the normal case for samples in radioactive disequilibria. In order to give a final answer, further detailed and systematic investigations are required, including the question whether the magnitude of radioactive disequilibria is a decisive factor for the $\mu$Dose-system's capability to determine values for the radionuclide concentrations that are in good agreement with results obtained by other methodological approaches. Regardless of the final answer to this question, we would like to point out that dose rates calculated from radionuclide concentrations of samples for which radioactive disequilibria have to be assumed will never be an accurate measure for trapped charge dating and should therefore be treated with care.

The overall good performance of $\mu$Dose-measurements is confirmed by the rate of agreement observed for the simulated environmental dose rates illustrated in Fig. 10. As described for the Nussy and Volkegem loess standards (see Sec. 5.1), these dose rates were calculated for the coarse ($90 - 200\,\mu$m) grain fraction of HF-etched quartz using DRAC v1.2 (Durcan et al., 2015). For calculation, we applied the conversion factors provided by Guérin et al. (2011) and used a constant water content of $15 \pm 5\,\%$ as well as a constant cosmic radiation of $0.150 \pm 0.015\,\mathrm{Gy \cdot ka^{-1}}$. We would like to point out that these values were arbitrarily chosen and do not represent the actual moisture and cosmic radiation values that might be detected for the different sampling locations.

Figure 10a shows a comparison of $\mu$Dose-based simulated environmental dose rates to values derived from TSAC/ICP-OES measurements performed at the Bayreuth Luminescence Laboratory. With samples Gi343, Gi455 and Gi465, there are three samples for which neither an agreement on the $1\sigma$-level nor on the $2\sigma$-level could be achieved. These samples have already been identified to be problematic (see discussion above). With respect to the Bayreuth samples, $88\,\%$ of the simulated environmental dose rates coincide within $2\sigma$, and still $79\,\%$ within $1\sigma$.

Figure 10b-d illustrate the results of $\mu$Dose-HRGS comparisons for different laboratories. With $25\,\%$ (Heidelberg), $36\,\%$ (Gliwice) and $50\,\%$ (Cologne), the proportions of samples for which an agreement on the $1\sigma$-level can be observed is substantially lower than for the $\mu$Dose-TSAC/ICP-OES comparison. On the $2\sigma$-level of agreement, $100\,\%$ (Heidelberg), $86\,\%$ (Gliwice) and $80\,\%$ (Cologne) of the calculated simulated dose rates coincide with the respective dose rate values derived from $\mu$Dose-measurements.

Overall, $55\,\%$ of the simulated environmental dose rates for all investigated samples coincide within $1\sigma$ and $88\,\%$ show an agreement on the $2\sigma$-level. In total, the measured-to-given-ratios range from $0.48$ to $2.81$ and average at a value of $1.04$, which improves to $1.00$ if the above mentioned three problematic samples are not considered. $80\,\%$ of the calculated measured-to-given ratios lie within $15\,\%$ of unity, indicating an overall very good rate of agreement for the simulated environmental dose rates. In summary, we can conclude that $\mu$Dose-measurements provide results which allow the calculation of dose rates that are in accordance with dose rate values derived from well-established methods of environmental dose rate determination.

Our findings do not point to significantly differing results for samples from different sedimentary environments. For aeolian sediments, $2\sigma$-levels of agreement of $80\,\%$ for uranium as well as $90\,\%$ for thorium and potassium were determined. For samples originating from fluvial environments, only ~$68\,\%$ of the uranium measurements agree on the $2\sigma$-level, what is slightly lower than for the aeolian sediments and might be attributed to potential radioactive disequilibria (see discussion above) or to a

**Table 10.** Proportions of agreement on the $2\sigma$-level between $\mu$Dose-results and results obtained by different techniques of determining radionuclide concentrations and/or activities (TSAC/ICP-OES and low-level HRGS). Results are grouped according to different sedimentary environments.

| Environmental setting | Proportion of agreement on the $2\sigma$-level | | |
|---|---|---|---|
| | Potassium | Thorium | Uranium |
| Aeolian sediments | 90 % | 90 % | 80 % |
| Fluvial sediments | 84 % | 89 % | 68 % |
| Littoral sediments | 33 % | 100 % | 100 % |
| Hillslope sediments | 75 % | 100 % | 75 % |
| Colluvial sediments | 86 % | 57 % | 71 % |
| Soil samples | 100 % | 67 % | 100 % |

stronger heterogeneity of the mineralogical composition of the fluvial deposits. With respect to thorium (89 %) and potassium (84 %), however, no significant differences between fluvial and aeolian samples were observed. Similar results were derived for

littoral samples as well as for hillslope sediments and soils (see summary in Table 10). With respect to colluvial samples, our findings at a first glance seem to point to slightly worse $2\sigma$-proportions of agreement for thorium (57 %) and uranium (71 %). A closer look at the results, however, shows that only seven colluvial samples were considered for this study. Two of them clearly revealed features of changing ground water levels and thus might most probably exhibit radioactive disequilibria. For at least three more samples, such disequilibria are likely if considering their sampling positions. Thus, due to the very specific

conditions at the respective sampling locations the colluvial samples investigated in this study proved to be problematic. Yet, we would like to emphasize that this result should not be generalized for colluvial samples. As a result, our study does not give evidence that samples from particular sedimentary environments are generally not suitable for analyses with the novel $\mu$Dose-system and should therefore a priori be excluded from $\mu$Dose-analyses. However, we would like to emphasize that the specific on-site conditions at the sampling locations are of decisive importance. The $\mu$Dose-system will only provide reliable results

for radionuclide concentrations if the fundamental requirement of secular equilibrium is met. Thus, a careful documentation of sampling locations, inter alia comprising sedimentological and hydrographic aspects, is indispensable for providing the database for a convincing interpretation of $\mu$Dose-results.

## 6 Conclusions

The $\mu$Dose-system is an easy to handle device that provides the possibility of determining the sample-specific concentrations

of uranium, thorium and potassium. Equipped with a dual layer scintillator sensitive to $\alpha$- and $\beta$-radiation, the system is able to discriminate between $\alpha$- and $\beta$-particles interacting with the scintillator and thus determine the total $\alpha$- and $\beta$-counts. Based on four decay pairs comprising two $\alpha$-$\alpha$-pairs and two $\beta$-$\alpha$-pairs, the measurement system allows discriminating series-specific activities, arising from the decay chains of $^{238}$U, $^{235}$U and $^{232}$Th. Based on the assumption that $^{40}$K is the dominant $\beta$-emitter

in natural samples that is not part of the above mentioned decay series, the $^{40}$K activity is calculated as a residual value derived from the excess of actually detected $\beta$-counts over the number of $\beta$-counts expected to arise from the sample-specific decay series of $^{238}$U, $^{235}$U and $^{232}$Th.

The results obtained with the $\mu$Dose-system are provided as activities ($\mathrm{Bq \cdot kg^{-1}}$) and as concentration values ($\mathrm{mg \cdot kg^{-1}}$ for U and Th; % for K). The results are summarized in the $\mu$Dose-software and in a dedicated report file. For user convenience, results and additional information are also exported to spreadsheet files that can easily be used as input files for various software solutions aiming at the calculation of sample-specific dose rates , such as laboratory-specific spreadsheets, R-based solutions, DRAC (Durcan et al., 2015), ADELE (Kuhlig, 2005) and others. However, the $\mu$Dose-system also provides the possibility to use an integrated dose rate calculation module, which was not considered for this study. Unlike other software solutions, the algorithms of this module consider the fact that uncertainties arising from the $\mu$Dose measurement process are correlated, which allows a significant improvement of dose rate precision.

In contrast to thick source alpha counting, the $\mu$Dose-system does not need any accompanying measurement procedures (e.g., ICP-OES, ICP-MS) in order to determine the potassium content. Compared to low-level HRGS, the new approach offers the advantage that it does neither require long storage times nor high technical efforts such as liquid nitrogen cooling or lead shielding.

The results of our performance test are quite promising. Our findings show that results gained by $\mu$Dose-measurements are characterized by an excellent or at least good reproducibility and that they reveal very good agreement with well-established dosimetry methods such as TSAC (in combination with ICP-MS or ICP-OES) and low-level HRGS.

Particularly for the certified IAEA standards, accuracy and reproducibility of the determined radioactivity values are excellent. This might be attributed to the high contents of radionuclides in these standards. For the loess standards, the reproducibility and accuracy are still good, yet not as perfect as for the IAEA reference materials. However, it has to be kept in mind that the reference values for the loess standards were derived from a limited number of measurements that might have been affected by specific methodological problems. The deviating results obtained with the $\mu$Dose-systems do not exceed the limits reported by other studies focussing on dosimetry. This also applies to the re-measurements of radionuclide concentrations of natural samples. With respect to the analysed environmental samples, our findings indicate very good agreement with results obtained by well-established methods. Overall, 71 % (U), 77 % (Th) and 78 % (K) of the values determined by $\mu$Dose-measurements agree to the benchmarks derived from either TSAC and ICP-OES or low-level HRGS within the $2\sigma$-level. On the $1\sigma$-level the proportion of agreement is still ~41 % for U, 46 % for Th and 61 % for K. Outlier samples for which no satisfying agreement of methods could be determined might be explained by the presence of radioactive disequilibria. However, the exact extent of the impact of such disequilibria on the $\mu$Dose measurement is not clear and will require further systematic investigations.

From a practical point of view, the $\mu$Dose-device allows the fast and cost-effective one-step-determination of radionuclide concentrations required for dose rate calculation in trapped charge dating. The sample preparation is straightforward and our findings indicate that rather short measurement times of ~2-4 days are sufficient to provide reliable information on radionuclide concentrations for samples revealing average levels of environmental radioactivity. The total number of detected $\alpha$-counts should be used as indicator for an adequate measurement duration, applying a threshold value of ~2,500 counts.

In summary, the $\mu$Dose-system is a promising tool for measuring low level concentrations of radionuclides in samples from natural environments. It has the potential to become a standard method for dose rate determination in routine luminescence and electron spin resonance dating applications.

### Appendix A: Measurement configuration for comparison of natural samples

**MEASUREMENT CONFIGURATION APPLIED IN THE BAYREUTH LUMINESCENCE LABORATORY** For the determination of uranium and thorium concentrations, thick source alpha counting (TSAC) was used whereas the potassium content was determined by ICP-OES, using a Varian Vista-Pro$^{TM}$ system. TSAC measurements were performed on a Littlemore Low Level Alpha Counter 7286 equipped with four photomultiplier tubes. Sample preparation included drying the sample material in a drying chamber at 105°C for several days, homogenizing and finally pulverizing the material using a ball mill. To ensure the complete coverage of the ZnS : Ag scintillation screen, the sample material was placed and gently compacted in a gas tight sample carrier consisting of acrylic glass. Before starting the TSAC measurements, all samples were stored for at least four weeks in order to account for radon emanation due to the sample preparation procedure.

**MEASUREMENT CONFIGURATION APPLIED IN THE COLOGNE LUMINESCENCE LABORATORY** For samples provided by the Cologne Luminescence Laboratory, uranium, thorium and potassium contents were determined by low level HRGS, using i) an Ortec Coaxial Profile M7080-S GEM high-precision Germanium Gamma-Ray detector with 60 % relative efficiency and connected to a Dspec jr 2.0; and ii) a Canberra Coaxial Profile GC4040 Germanium Gamma-Ray detector with a relative efficiency of 20 % connected to an Ortec 92x Spectrum Master. Samples were dried at 50°C for at least two days, crushed in a jaw breaker if necessary and homogenized. Depending on the available amount of sample material, polypropylene (PP) capsules with calibrated capacities of 200 g and 590 g were filled to the top, tape sealed and stored for four weeks to compensate for radon loss induced by sample preparation. The capsules were then placed on top of the detector surrounded by a 10 cm thick lead shield and measured for 42 hours. GammaVision 8.0 software with the LVis 3.0.9 application was used for measurements and analyses. $^{40}$K activities were directly measured based on the gamma line at $1,461$ keV. $^{238}$U activities were derived from the gamma lines at 295 keV, 352 keV, 609 keV, $1,120$ keV, $1,764$ keV and $2,204$ keV. For determining $^{232}$Th activities, the following gamma lines were used: 209 keV, 338 keV, 911 keV, 965 keV, 969 keV, 727 keV, 583 keV, 861 keV and $2,614$ keV. A summarizing compilation of used gamma lines can be found in Table A1. Nussy loess was utilized for efficiency calibration of the individual sample containers, whereas $^{152}$Eu (50 kBq) and $^{60}$Co (37 kBq) check sources were used for periodic energy calibration and quality checks.

**MEASUREMENT CONFIGURATION APPLIED IN THE HEIDELBERG LUMINESCENCE LABORATORY** Radionuclide determination in Heidelberg was based on low level HRGS. The sample material was dried (50°C, few days until no further weight loss was observed), weakly pestled for homogenization and filled in a sealed plastic container (filling capacity approximately 30 g). In order to compensate for potential $^{222}$Rn loss during the preparation process, the samples were stored for at least four weeks. Thereafter, a lead shielded broad energy Ge detector (Canberra, model BE 2020) was used to determine the sample concentration of $^{238}$U, $^{232}$Th and $^{40}$K. While $^{40}$K could be measured directly, for $^{238}$U and $^{232}$Th, the gamma lines

**Table A1.** Compilation of radioactive daughter nuclides and their associated gamma peaks used for low-level HRGS-based uranium, thorium and potassium determination in the participating laboratories of Cologne, Heidelberg and Gliwice.

| Daughter nuclide | Gamma line at... | Cologne laboratory | Heidelberg laboratory | Gliwice laboratory |
|---|---|:---:|:---:|:---:|
| **$^{238}$U decay chain** | | | | |
| $^{234}$Th | 63 keV | | ■ | |
| $^{226}$Ra | 186 keV | | ■ | |
| $^{214}$Pb | 295 keV | ■ | ■ | ■ |
| | 352 keV | ■ | ■ | ■ |
| $^{214}$Bi | 609 keV | ■ | ■ | ■ |
| | 1,120 keV | ■ | ■ | ■ |
| | 1,764 keV | ■ | ■ | |
| | 2,204 keV | ■ | | |
| $^{210}$Pb | 47 keV | | ■ | |
| **$^{232}$Th decay chain** | | | | |
| $^{228}$Ac | 129 keV | | ■ | |
| | 209 keV | ■ | ■ | |
| | 338 keV | ■ | ■ | |
| | 911 keV | ■ | ■ | ■ |
| | 965 keV | ■ | | |
| | 969 keV | ■ | ■ | |
| $^{212}$Pb | 239 keV | | ■ | |
| $^{212}$Bi | 727 keV | ■ | | |
| $^{208}$Tl | 583 keV | ■ | ■ | ■ |
| | 861 keV | ■ | | |
| | 2,614 keV | ■ | ■ | ■ |
| **$^{40}$K decay** | | | | |
| $^{40}$K | 1,461 keV | ■ | ■ | ■ |

of their decay products ($^{234}$Th, $^{226}$Ra, $^{214}$Pb, $^{214}$Bi and $^{210}$Pb for $^{238}$U; $^{228}$Ac, $^{212}$Pb and $^{208}$Tl for $^{232}$Th) were measured and combined using a weighted mean. This allowed detecting possible radioactive disequilibria in the uranium chain, which, however, were not an issue for the Heidelberg samples presented here. A detailed overview of used gamma lines can be found in Table A1. Regular measurements of an identically treated standard (Kasper et al., 2001; Preusser and Kasper, 2001) were implemented to calibrate the detector and monitor its performance.

**MEASUREMENT CONFIGURATION APPLIED AT THE INSTITUTE OF PHYSICS IN GLIWICE** The decay chains of $^{238}$U and $^{232}$Th as well as $^{40}$K concentrations were measured by low-level HRGS using a HPGe detector (Canberra GX 4518) and Genie-PC software (Canberra). The investigated samples were stored in a laboratory dryer for a few days, depending on moisture. The dried samples were crushed and 100 g of each sample were sealed in gBeakers (Poręba et al., 2020). Prior to measurement, samples were stored for at least three weeks. This delay was necessary to allow $^{222}$Rn to reach a radioactive equilibrium with $^{226}$Ra. The measurement time for each sample was about 24 hours (Moska et al., 2021). To obtain the $^{238}$U content the following gamma lines were considered: 295 keV, 352 keV, 609 keV and 1,120 keV. To calculate the $^{232}$Th activity the following gamma lines were considered: 583 keV, 911 keV and 2,614 keV. For $^{40}$K the gamma line at 1,461 keV was used. For a summarizing compilation of used gamma lines, the reader is referred to Table A1. The HRGS-system was calibrated using the RGU-1, RGTh-1 and RGK-1 reference materials provided by the IAEA. Regularly applied quality controls are implemented in the measurement routines in Gliwice using reference material IAEA-385.

**Appendix B: Overview of natural samples**

**Table B1.** Compilation of 47 natural samples investigated for this study. These samples have been provided by four different laboratories and represent various environmental settings. A more detailed description of sample characteristics, sampling locations and research contexts is given in Appendix C.

| Sample ID | Information on sampling location | | | Location name & country | Sediment characterization | Publications |
|---|---|---|---|---|---|---|
| | Latitude | Longitude | Elevation | | | |
| **Cologne Luminescence Laboratory** | | | | | | |
| Col_GGW1 | 50.980°N | 7.160°E | 135 m a.s.l. | Paffrather Mulde (Germany) | Colluvium | Zander et al. (2019) |
| Col_GGW2 | 50.980°N | 7.160°E | 135 m a.s.l. | Paffrather Mulde (Germany) | Colluvium | Zander et al. (2019) |
| Col_GGW3 | 50.980°N | 7.160°E | 135 m a.s.l. | Paffrather Mulde (Germany) | Colluvium | Zander et al. (2019) |
| Col_GGW4 | 45.760°N | 4.840°E | 180 m a.s.l. | Lyon (France) | Fluvial sands | - |
| Col_GGW5 | 44.337°N | 4.702°E | 50 m a.s.l. | Rhone Valley (France) | Fluvial sands | - |
| Col_GGW6 | 50.766°N | 13.716°E | 727 m a.s.l. | Rote Weißeritz (Germany) | Alluvium | Tolksdorf et al. (2020) |
| Col_UGW1 | 22.300°S | 114.15°E | 4 m a.s.l. | Point Lefroy (Australia) | Littoral sands | Brill et al. (2017); May et al. (2017) |
| Col_UGW2 | 22.300°S | 114.15°E | 4 m a.s.l. | Point Lefroy (Australia) | Littoral sands | Brill et al. (2017); May et al. (2017) |
| Col_UGW3 | 22.300°S | 114.15°E | 4 m a.s.l. | Point Lefroy (Australia) | Littoral sands | Brill et al. (2017); May et al. (2017) |
| Col_UGW4 | 33.540°N | 9.950°E | 351 m a.s.l. | Matmata Plateau (Tunesia) | Loess | Faust et al. (2020) |
| **Giessen Luminescence Laboratory** | | | | | | |
| Gi142 | 50.450°N | 8.770°E | 198 m a.s.l. | Münzenberg (Germany) | Loess | Lomax et al. (2018) |
| Gi263 | 49.015°N | 12.096°E | 332 m a.s.l. | Regensburg (Germany) | Alluvium | - |
| Gi311 | 48.092°N | 8.1653°E | 1022 m a.s.l. | Black Forrest (Germany) | Colluvium | Henkner et al. (2017); Miera et al. (2019) |
| Gi324 | 45.338°N | 97.912°E | 2343 m a.s.l. | Western-Bogd-Fault (Mongolia) | Aeolian silt | Ritz et al. (1995) |
| Gi325 | 45.338°N | 97.912°E | 2343 m a.s.l. | Western-Bogd-Fault (Mongolia) | Fan/river deposit | Ritz et al. (1995) |
| Gi329 | 22.897°S | 64.675°W | 773 m a.s.l. | Rio Iruya (Argentina) | Fluvial terrace | - |
| Gi335 | 28.660°N | 13.870°W | 130 m a.s.l. | Fuerteventura Island (Spain) | Stone pavement | Fuchs and Lomax (2019) |
| Gi338 | 28.650°N | 13.850°W | 82 m a.s.l. | Fuerteventura Island (Spain) | Stone pavement | Fuchs and Lomax (2019) |
| Gi341 | 37.809°S | 73.014°W | 1200 m a.s.l. | Agnol (Chile) | Hillslope sediment | - |
| Gi343 | 50.040°N | 11.230°E | 457 m a.s.l. | Weismain (Germany) | Colluvium | - |

| Sample ID | Information on sampling location | | | Location name & country | Sediment characterization | Publications |
|---|---|---|---|---|---|---|
| | Latitude | Longitude | Elevation | | | |
| Gi344 | 50.040°N | 11.230°E | 457 m a.s.l. | Weismain (Germany) | Colluvium | - |
| Gi354 | 35.240°N | 116.05°W | 302 m a.s.l. | Mojave Desert (USA) | Stone pavement | Bateman et al. (2012) |
| Gi360 | 47.886°N | 91.415°E | 1640 m a.s.l. | Hovd Fault Zone (Mongolia) | Fluvial sand | Rogozhin et al. (2013) |
| Gi433 | 26.129°S | 70.525°W | 475 m a.s.l. | Pan de Azucar (Chile) | Hillslope sediment | - |
| Gi434 | 26.129°S | 70.525°W | 475 m a.s.l. | Pan de Azucar (Chile) | Hillslope sediment | - |
| Gi437 | 26.127°S | 70.529°W | 456 m a.s.l. | Pan de Azucar (Chile) | Hillslope sediment | - |
| Gi450 | 50.750°N | 8.730°E | 173 m a.s.l. | Niederweimar (Germany) | Fluvial terrace | Lomax et al. (2018) |
| Gi451 | 50.750°N | 8.730°E | 173 m a.s.l. | Niederweimar (Germany) | Fluvial terrace | Lomax et al. (2018) |
| Gi452 | 50.750°N | 8.730°E | 173 m a.s.l. | Niederweimar (Germany) | Fluvial terrace | Lomax et al. (2018) |
| Gi453 | 50.750°N | 8.730°E | 173 m a.s.l. | Niederweimar (Germany) | Fluvial terrace | Lomax et al. (2018) |
| Gi455 | 50.750°N | 8.730°E | 173 m a.s.l. | Niederweimar (Germany) | Floodplain loam | Lomax et al. (2018) |
| Gi465 | 50.730°N | 8.710°E | 172 m a.s.l. | Niederwalgern (Germany) | Alluvium | Lomax et al. (2018) |
| Gi466 | 50.730°N | 8.710°E | 172 m a.s.l. | Niederwalgern (Germany) | Alluvium | Lomax et al. (2018) |
| Gi649 | 50.730°N | 8.710°E | 172 m a.s.l. | Niederwalgern (Germany) | Alluvium | Lomax et al. (2018) |

**Heidelberg Luminescence Laboratory**

| Sample ID | Latitude | Longitude | Elevation | Location name & country | Sediment characterization | Publications |
|---|---|---|---|---|---|---|
| HDS-1378 | 3.895°N | 12.070°E | 703 m a.s.l. | Southern Cameroon Plateau | Tropical soil | - |
| HDS-1381 | 3.895°N | 12.070°E | 703 m a.s.l. | Southern Cameroon Plateau | Tropical soil | - |
| HDS-1386 | 3.873°N | 12.270°E | 711 m a.s.l. | Southern Cameroon Plateau | Tropical soil | - |
| HDS-1726 | 50.025°N | 104.99°W | 590 m a.s.l. | Avonlea Badlands (Canada) | Silt loam deposit | Hardenbicker and Bitter (2017) |
| HDS-1742 | 49.853°N | 8.772°E | 200 m a.s.l. | Messel uplands (Germany) | Aeolian sands | - |
| HDS-1744 | 49.853°N | 8.772°E | 200 m a.s.l. | Messel uplands (Germany) | Aeolian sands | - |
| HDS-1763 | 49.822°N | 8.822°E | 174 m a.s.l. | Reinheim (Germany) | Fossil soil | Semmel (1974) |
| HDS-1767 | 49.822°N | 8.822°E | 174 m a.s.l. | Reinheim (Germany) | Fossil soil | Semmel (1974) |

**Institute of Physics (Gliwice)**

| Sample ID | Latitude | Longitude | Elevation | Location name & country | Sediment characterization | Publications |
|---|---|---|---|---|---|---|
| CD_15 | 51.348°N | 22.094°E | 180 m a.s.l. | Kazimierz Dolny (Poland) | Colluvium | - |
| CD_16 | 51.348°N | 22.094°E | 180 m a.s.l. | Kazimierz Dolny (Poland) | Colluvium | - |
| DJ_56 | 53.643°N | 18.165°E | 95 m a.s.l. | Grudziądz (Poland) | Fluvial sands | Rurek et al. (2016) |
| U_1_2 | 50.390°N | 18.380°E | 214 m a.s.l. | Ujazd (Poland) | Colluvium | Jersak (1973) |
| U_1_19 | 50.390°N | 18.380°E | 214 m a.s.l. | Ujazd (Poland) | Colluvium | Jersak (1973) |

**Appendix C:  Detailed description of natural samples analysed for this study**

For this study, numerous natural samples representing a great variety of environmental settings from all over the world have been considered. While Table B1 in Appendix B gives a short summary of all investigated samples focussing on the very basic facts, the following sections in Appendix C will provide more detailed sample characterizations as well as concise descriptions of sampling locations, lithologies and research contexts.

**C1    Samples provided by the Giessen Luminescence Laboratory (Germany)**

**Gi450-Gi453, Gi455, Gi465-Gi466 and Gi649** – *Fluvial sediments from terrace gravels and floodplain loams in Germany*
Gi450-Gi453, Gi455, Gi465-Gi466 and Gi649 originate from two gravel quarries located in the surroundings of the city of Giessen, Germany. Samples Gi450-Gi453 and Gi455 were collected from the lower terrace of the Lahn river in the gravel quarry at Niederweimar (50.75°N, 8.73°E, 173 m a.s.l.), which is located in the central Lahn valley some kilometres south of the city of Marburg, Germany. With the Lahn river cutting through various geological units, the composition of the gravel

spectrum is rather versatile with dominant contributions of greywacke and sandstones, associated with radiolarites, basalt and quartzites (Lomax et al., 2018). The basement of the gravel deposits is built-up of Upper Permian sandstones and claystones (Zechstein formation). Revealing at least three distinct units, the fluvial gravels show a total thickness of $8 - 10$ m and are covered by $3 - 4$ m of late Pleistocene and early Holocene cover sediments (Lomax et al., 2018). Despite sample Gi455, which was taken from the overlying floodplain loams, all samples originate from either Unit II or Unit III of the fluvial gravel deposits

which are characterized by a compact body of medium to coarse gravels embedded in a sandy matrix and interstratified with several sand lenses (Lomax et al., 2018).

Samples Gi465, Gi466 and Gi649 were taken in the former gravel quarry of Niederwalgern, Germany (50.73°N, 8.71°E, 172 m a.s.l.). The samples originate from Holocene alluvial sediments covering Late Pleistocene fluvial gravels of the lower terrace of the Lahn river. Characterized by a predominant amount of silt and revealing numerous pieces of charcoal and ceramic

fragments, these alluvial sediments have been OSL-dated to the medieval period around $1$ ka (Lomax et al., 2018). For a detailed description of the litho- and biostratigraphic characteristics of the location the reader is kindly referred to Urz (1995).
**Gi335, Gi338 and Gi354** – *Samples from stone pavement areas*
Samples Gi335 and Gi338 were collected from fine grain material underlying stone pavement layers in the northern part of Fuerteventura Island (Canary Islands, Spain). They originate from two different profiles (Gi335: 28.66°N, 13.87°W, 130 m

a.s.l.; Gi338: 28.65°N, 13.85°W, 82 m a.s.l.) situated on a Middle Pleistocene basaltic lava flow (Fuchs and Lomax, 2019).

Sample Gi354 was taken at a location on the Soda Lake Sand Ramp in the Mojave Desert, California (USA, 35.24°N, 116.05°W, 302 m a.s.l.). The sand ramp is covered by a thin layer of coarse clasts forming a typical desert pavement surface. For a detailed description of the surrounding area as well as of the geological setting, the reader is kindly referred to Bateman et al. (2012), who are discussing the formation of sand ramps based on a detailed investigation of a nearby sand ramp at Soldier

Mountain (Mojave Desert, California).

**Gi142** – *Loess sample from the Münzenberg loess section*

The loess section at Münzenberg (50.45°N, 8.77°E, 198 m a.s.l.) is located in the northern part of the Wetterau area, a loess area in the south-western part of Hesse, Germany. Flanked by the Miocene Vogelsberg basaltic complex to the east and the Taunus mountains to the west, the Northern Wetterau area is part of the Hessian Depression. It is characterized by a gently rolling landscape, developed on widely unconsolidated Upper Tertiary sedimentary rocks associated with deeply weathered Miocene basalts (Lomax et al., 2018). Throughout the Pleistocene period, the whole region was a zone of loess accumulation. In sheltered positions, loess deposits have been preserved, frequently revealing thicknesses of more than 10 m (Schönhals, 1996). Taken at a depth of ~8 m, the investigated sample Gi142 originates from the lower part of the section, representing material for which a pre-Eemian age (pre-MIS5e) was determined (Lomax et al., 2018).

**Gi263, Gi311, Gi343 and Gi344** – *Colluvial sediments and archaeological sites from Southern Germany*

Sample Gi263 is part of a set of luminescence samples that were taken during archaeological excavations in the medieval city centre of Regensburg (Germany, 49.015°N, 12.096°E, 332 m a.s.l.). Situated on the southern bank of the Danube river, the sample consists of fine-grained fluvial material.

Sample Gi311 originates from an archaeological site in the south-eastern part of the central Black Forrest (SW Germany). The sample was collected from colluvial sediments in the upper reaches of a small valley close to the origin of the Breg river (48.0919°N, 8.1653°E, 1,022 m a.s.l.), the longest headwater stream of the Danube river. Generally characterized by deeply incised valleys with steep slopes and revealing elevations of up to 1,100 m a.s.l., the lithology of the sampling site is dominated by crystalline rock formations of the Variscan basement. A detailed description of the geological and geomorphological setting of the whole region and its relevance for the Neolithic settlement dynamics is given by Henkner et al. (2017) and Miera et al. (2019).

Samples Gi343 and Gi344 originate from loess bearing colluvial sediments in the catchment area of the river Weismain (Upper Franconia, Germany; 50.04°N, 11.23°E, 457 m a.s.l.). The lithology of the area is dominated by Mesozoic limestone formations and dolomites covered by sporadic loess loam layers. As part of an archaeological excavation, the samples were taken in the vicinity of a former human settlement attributed to the Urnfield period.

**Gi324, Gi325, Gi329, Gi341, Gi433-Gi434, Gi437, Gi360** – *various samples from high mountain areas*

Samples Gi324 and Gi325 originate from the Western-Bogd-Fault, a still active tectonic fault system in the south-western part of Mongolia. Both samples were taken at depths of ~40 m b.g.l. and ~60 m b.g.l. from fine-grained sediments made accessible by two trenches crossing the fault system (45.3375°N, 97.9118 °E, 2,343 m a.s.l.). Gi324 represents surface deposits consisting of un-stratified aeolian silts showing distinct features of bioturbation and cryoturbation. Gi325 was collected from stratified fan and river deposits characterized by an alternating sequence of gravels, sands and intercalated silty to clayish layers. For a detailed description of the fault system and its surroundings, the reader is kindly referred to Ritz et al. (1995). A characterization of the regional tectonic and geological setting can be found in Rizza et al. (2011).

Like samples Gi324 and Gi325, sample Gi360 also originates from the Mongolian Altay mountains, however approximately 300 km further to the north. It was collected at a depth of ~5 m b.g.l. from a sand pocket within a fluvial terrace in a small

valley of the Hovd Fault Zone (47.8859°N, 91.4145°E, 1640 m a.s.l.). The tectonic setting of the Hovd Fault area is described inter alia by Rogozhin et al. (2013).

Sample Gi329 was taken from an alluvial terrace located in the valley of the river Rio Iruya in the north-western part of Argentina (22.89677°S, 64.67518°W, 773 m a.s.l.). The terrace is built-up of well-rounded, coarse clastic gravels and boulders embedded into a matrix of silty sand and shows a total thickness of several meters. The coarse fluvial sediments are divided into distinct sub-units by several layers of fine-grained material. At least one palaeosol horizon was identified. Sample Gi329 originates from a layer of silty material below this palaeosol horizon and was $^{14}$C-dated to approximately 17.6 ka.

Samples Gi341, Gi433, Gi434 and Gi437 were collected at various locations of the Chilean coastal cordillera. Sample Gi341 represents hillslope sediments originating from a northern-facing slope (37.809°S, 73.0136°W, 1200 m a.s.l.) located in the Parque Nacional Nahuelbuta, a national-park approximately 30 km west of the town of Agnol (southern Chile). Like other areas of the Nahuelbuta Mountains, the geology of the location is dominated by quartz-rich granites of Late Palaeozoic age. As part of a Cambisol (IUSS Working Group WRB, 2007), sample Gi341 was taken at a depth of ~70 cm b.g.l. from a transition zone between a layer of clastic boulders and a cambic horizon developed on deeply weathered saprolite.

Samples Gi433, Gi434 and Gi437 were taken from two outcrops in the Pan de Azucar, which is part of the Atacama Desert in northern Chile. At the sampling site, several distinct surface levels representing different generations of alluvial fans can be distinguished. Gi433 and Gi434 were collected from level 2 of this fan system at depths of ~25 cm b.g.l. and ~90 cm b.g.l., respectively. The outcrop (26.129°S, 70.525°W, 475 m a.s.l.) was characterized by an alternating sequence of coarse detritus material and layers of medium to coarse sand. Sample Gi437 represents material of level 3 of the alluvial fan system. Like the other two samples, it was taken at a depth of ~30 cm b.g.l. from an alternating sequence of coarse and fine materials accumulated in an adjacent small valley (26.127°S, 70.529°W, 456 m a.s.l.).

## C2   Samples provided by the Heidelberg Luminescence Laboratory (Germany)

**HDS-1378, HDS-1381 and HDS-1386** – *Ferralsol soil samples from Southern Cameroon*

HDS-1378, HDS-1381 and HDS-1386 are from two sites in a tropical, semi-deciduous rainforest area on the Southern Cameroon Plateau at about 700 m a.s.l. The profiles were dug into outcrops along the national road N10 between Yaoundé and Bertoua, approximately 10.5 km NE (site AK-R: HDS-1386) and 23.5 km NW (site AK-Y: HDS-1378, HDS-1381) linear distance from Akonolinga. Both sites show deeply weathered soils of the Ferralsol type (FAO, 2006), with pisoplinthic horizons in $3 - 5$ m depth. The ferralic horizons consist mainly of quartz, kaolinite and iron-oxides (total Fe $5 - 6$ %). AK-Y has a clay texture, with little silt, and sand contents of $30 - 35$ %, whereas AK-R is a sandy clay, with sand contents of $60 - 65$ %. The yellowish-brown hue (7.5YR) in the top meter of AK-Y, compared to the red hue (2.5YR) in its subsoil and throughout AK-R may be the result of higher moisture content due to impeded drainage and related xanthisation, resulting in the formation of goethite (yellow) rather than hematite (red; cf. Cornell and Schwertmann, 2003). Both sites are supposed to be subjected to intensive bioturbation, especially due to termite activity. The samples HDS-1378 and HDS-1381 were taken from AK-Y at 190 cm and 70 cm depth, respectively; HDS-1386 from AK-R was collected from 360 cm depth, where the occurrence of fine gravel was noted.

**HDS-1726** - *Avonlea Badlands in Canada*

Sample HDS-1726 is from the semi-arid Canadian Prairies in southern Saskatchewan, approximately 55 km linear distance SW of Regina, 5 km NE of the village Avonlea and 200 m west of the extensively meandering Avonlea Creek. After the recession of the Wisconsian ice-sheet, fluvial incision of a glacial meltwater channel in a sequence of Upper Cretaceous sandstones, mudrocks and bentonite initiated the evolution of the Avonlea Badlands. Nowadays, this region can be characterized as a typical badland with high erosion rates from overland and pipe flow (Hardenbicker and Bitter, 2017). HDS-1726 represents a modern, light coloured silt loam deposit from 10 cm b.g.l. from the lower pediment of the study site (cf. Fig. 3 in Hardenbicker and Bitter, 2017).

**HDS-1742 and HDS-1744** – *Aeolian cover sands from the Messel uplands*

Samples HDS-1742 and HDS-1744 are from an abandoned sandpit near Roßdorf in the Messel uplands, continuing the Oden-wald mountains to the north, approximately 7.5 km east of the city of Darmstadt in southern Hesse, Germany. Rotliegend mudstones are covered by Pleistocene cover sands and dunes. In the south-eastern direction the sandy deposits grade into loess and sandy loess deposits (cf. samples HDS-1763 and HDS-1767). Garnet and epidote are dominant heavy minerals pointing to the Upper Rhein Graben to the west as the source area of the aeolian sands. HDS-1742 was collected from a fossil, humic top horizon showing secondary carbonate precipitation at 185 cm b.g.l. while HDS-1744 was taken from an aeolian layer at 140 cm b.g.l., which was situated below the remains of a truncated Holocene Luvisol. Whereas the upper sample represents a sand ($< 10$ wgt $- \%$ silt and clay), the lower sample is a sandy loam with ~67 wgt $- \%$ sand, ~28 wgt $- \%$ silt and ~5 wgt $- \%$ clay, suggesting that at the time of sediment accumulation the transition from sand to sandy loess deposits was further to the west.

**HDS-1763 and HDS-1767** – *Loess section in the former brickyard "Grün" near Reinheim, Hesse*

Samples HDS-1763 and HDS-1767 are from a loess-palaeosol section near Reinheim, approximately 14 km southeast of Darmstadt, along the Wembach at the northern rim of the Odenwald Mountains in southern Hesse, Germany. On top of pre-Quaternary clays and $1 - 2$ m of fluvial gravel likely from an early-Pleistocene fluvial terrace, up to 20 m of loess were exposed (Semmel, 1974) at times when the site was used by the former brickyard "Grün" for the extraction of loam (in operation 1872 – 2013). Apart from one to three fossil Stagnosol (Sd; Ad-Hoc-AG Boden, 2005) horizon(s) in the basal part, up to five fossil clay-illuviation (Bt) horizons were observed, two of them above the Reinheim tephra (Semmel, 1967, 1995). After refilling of the loam pit, approximately 9.5 m of the loess section are still accessible, exhibiting one pronounced clay-illuviation horizon ($4.45 - 5.5$ m b.g.l.) above the Reinheim tephra (~9 m b.g.l.) (Anefeld et al., 2018). Sample HDS-1763 was taken from a fossil leached, stagnic (fAl-Ssw) horizon at 4.3 m b.g.l., right on top of the fBt-Sd. The soil material showed plenty of charcoal pieces, likely as a consequence of wild fires, and gave indication of soil reworking. Sample HDS-1767 was taken at 3.1 m b.g.l. from a fossil horizon showing strong secondary carbonate precipitation (Ckc; concretions up to 15 cm diameter). Both samples represent silty clay loam and showed pH-values of ~7.7. Based on luminescence dating ($pIR_{60}IR_{225}$ SAR protocol), ages of $226 \pm 18$ ka for HDS-1767 and $221 \pm 15$ ka for HDS-1763 could be determined.

## C3 Samples provided by the Cologne Luminescence Laboratory (Germany)

The Cologne Luminescence Laboratory overall provided ten samples which were subject of different research projects including littoral environments and geo-archaeological settings as well as alluvial sediments and aeolian deposits.

**Samples Col_GGW1 – Col_GGW3** were taken during an archaeological excavation of a Roman lime kiln situated on the western slope of a small hill within the Paffrather Mulde (50.98°N, 7.16°E, 135 m a.s.l.) near the city of Bergisch Gladbach (Germany). The local lithology is dominated by Devonian limestone and dolomite covered by silty weathered loam. Samples Col_GGW1 and Col_GGW2 were extracted from a fritted contact zone between the packing chamber and the surrounding sediments. Col_GGW3 originates from an oxidized, reddish-brown residual loam outside the contact area. For details, the
reader is kindly referred to Zander et al. (2019).

**Samples Col_GGW4 – Col_GGW6** represent fluvial environments. While Col_GGW4 was taken during an archaeological excavation in the city centre of Lyon (France, 45.76°N, 4.84°E, 180 m a.s.l.), Col_GGW5 originates from the alluvial plain of the Rhone river near the town of Pierrelatte (France, 44.337°N, 4.702°E, 50 m a.s.l.). Both samples were taken from fluvial sands of alluvial deposits accumulated by the Rhone river. Sample Col_GGW6 originates from alluvial sediments of the
865 Rote Weißeritz river near the town of Schellerhau (Erzgebirge Mountains, Germany, 50.766°N, 13.716°E, 727 m a.s.l.). For a detailed description of the sampling location the reader is kindly referred to Tolksdorf et al. (2020).

**Samples Col_UGW1 – Col_UGW3** originate from a littoral environment. They have originally been analysed as part of the investigation of washover fans at Point Lefroy (22.30°S, 114.15°E, 4 m a.s.l.), which is located in the Exmouth Gulf in the north-western part of Western Australia. All samples have been taken from littoral sandy deposits consisting of a mixture
of siliciclastic sand, coral fragments and shells. A detailed description of the sampling location including the geological and geomorphologic settings as well as a thorough sedimentary characterization are given by Brill et al. (2017) and May et al. (2017).

Finally, **Col_UGW4** is a loess sample from the Matmata Plateau (Tunesia). The lithology of the plateau is dominated by mid-Cretaceous limestones showing several basins filled with sandy loess deposits. The sample was taken near the village of
875 Matmata (33.54°N, 9.95°E, 351 m a.s.l.). A detailed description of the Matmata loess region is given by Faust et al. (2020).

## C4 Samples provided by the Institute of Physics in Gliwice (Poland)

**U_1_2 and U_1_19** – were collected for studies on soil erosion and sedimentation processes applying fallout radionuclides. Both samples were collected from an agricultural field located on a gentle slope within the Proboszczowicki tableland near the village of Ujazd (South Poland). The samples are colluvial sediments and were collected at the base of the slope. The
880 sampling site is located in an area overall characterized by Pleistocene loess sediments that were described as "transition loess formation" by Jersak (1973). While the mean grain size of sample U_1_2 is equal to $40\,\mu m$ (very coarse silt), it is about $139\,\mu m$ for sample U_1_19 (very fine sand).

**CD_15 and CD_16** – were collected near the town of Kazimierz Dolny, which is located on the Nałęczów Plateau (East Poland). The samples were part of a research project dealing with Holocene transformation of loess areas. The samples originate from colluvial sediments filling a fossil gully. These deposits were strongly modified by pedogenic processes.

**DJ_56** is a sample collected from fluvial sediments in a small valley near the city of Grudziądz in the northern part of Poland. This sample represents layered fluvial sands revealing a mean grain size of about 280 μm. Further information is provided by Rurek et al. (2016).

## Appendix D: Total alpha counts and measurement durations for Nussy and Volkegem loess standards

The following table shows the relation of total alpha counts to measurement durations for the analysed loess standards Nussy and Volkegem. The given values represent average values derived from measurements performed on three different devices. The values provided in the table might be used as a rule of thumb to roughly estimate required measurement durations for natural samples. However, we would like to point out that the time necessary to reach a particular alpha count level will not only depend on the dose rate of the analysed sample, but also on the sample-specific composition of radionuclides. For instance, a sample revealing a low dose rate due to low $^{40}$K-activity may still have rather high uranium and thorium contents. Such a sample can reach the 2,500 $\alpha$-count level much faster than a high dose rate sample with extremely high K-content but very low U- and Th-concentrations. Furthermore, it should also be considered that the required measurement duration will also depend on the device-specific settings.

**Table D1.** Relation of total alpha counts to measurement duration for the Nussy and Volkegem loess standards. Given values represent mean values derived from all measurements on three different μDose-devices.

| Total $\alpha$-counts | Nussy loess standard (1.93 ± 0.07 Gy·kg$^{-1}$) | Volkegem loess standard (2.71 ± 0.15 Gy·kg$^{-1}$) |
|---|---|---|
| $< 500$ | 12 h | NA |
| $500 - 1000$ | 26 h | 26 h |
| $1000 - 2000$ | 67 h | 72 h |
| $2000 - 3000$ | 91 h | 77 h |
| $> 3000$ | 170 h | 135 h |

*Data availability.* Data used for calculations in this paper are either summarized in the tables of the paper or stored as .csv- and .xlsx-files on the Research Data Repository "JLUdata" provided by the Justus Liebig University Giessen. These additional data are available at http://dx.doi.org/10.22029/jlupub-39.

*Author contributions.* TK conceived the study with important input from KT, JL and MF. AK, GP, AZ and LZ carried out the low-level HRGS measurements for this study. TK did parts of the TSAC measurements and carried out $\mu$Dose-measurements at the Giessen Luminescence Laboratory. Supported by JL and MF, TK compiled and analysed the data. TK wrote the manuscript with significant contributions from all co-authors.

*Competing interests.* The authors declare that they have no conflict of interest.

*Acknowledgements.* The authors would like to thank Dimitri Vandenberghe for providing the Volkegem loess standard, Manfred Fischer performing and organising the TSAC and ICP-OES measurements at the University of Bayreuth and Matthias Schick, who was in charge of the sample preparation for $\mu$Dose-measurements at the Giessen Luminescence Laboratory. Furthermore, the authors thank Ulrike Hardenbicker, Tobias Sprafke and Heinrich Thiemeyer for relevant information on the presented samples from the Heidelberg Luminescence Laboratory. Finally, we would like to thank the two anonymous reviewers for their constructive and helpful comments that allowed us to significantly improve our manuscript.

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
