# Peer review of "The $\mu$ Dose-system: determination of environmental dose rates by combined alpha and beta counting – performance tests and practical experiences"

_Geochronology, 2021_

## Author Comment (AC1)

**Reply to Reviewer 1**

We thank Reviewer 1 for his/her constructive and helpful comments, allowing us to improve our manuscript. Below, we provide a detailed response to the raised concerns and suggestions. For clarity, quotes from the review are italicized.

Specific comments:

- *Because of the algorithm used in uDose-system, the U and Th contents are negatively correlated. We can see from Figures 3, 4, 7, 8 that, when the Th content or activity is higher (assume it is overestimated), the corresponded U is lower (underestimated), and vice versa. I think it might be more helpful to add the bulk U+Th activity as another parameter for comparison. Even though the individual activities of Th and U are deviated from the expected values, as long as the bulk U+Th activity is close to the expected value, it might still be treated as a successful measurement regarding the calculation of environmental dose rate. The conversion factor from activity to beta dose rate is higher for U and lower for Th, while the conversion factor from activity to gamma dose rate is higher for Th and lower for U. They compensate each other, and the total dose rate does not vary much with the exact Th/U ratio (e.g. section 4.3.1 in Aitken 1985; Li and Tso, 1995). For example, in thick source alpha counting (TSAC), sometimes we can simply assume that the sample has equal activities of U and Th series. So, I think the bulk U+Th activity would be another evidence for the reliability of µDose-system to accurately determine the environmental dose rate.*

We thank Reviewer 1 for this interesting comment and added plots that show bulk U+Th activities of the investigated loess standards Nussy and Volkegem. These additional plots were combined with plots showing simulated dose rates calculated for the standards. Furthermore, the results shown in these plots are discussed in the manuscript.

- *Maybe, it would be even more straightforward to calculate the final environmental dose rates for comparison. Dose rates can be calculated according to the true settings of individual samples (grain size, mineral, water contents, etc). Alternatively, dose rates of all samples can be simply calculated based on etched quartz with a fixed diameter (e.g. 150-200 µm) by U, Th, K measured from µDose and other methods (TSAC, ICP-OES, gamma spectrometry), and just assume constant cosmic ray and water content. Comparison of dose rates can directly give the readers an impression about the performance of µDose in determining the environmental dose rate.*

Done. We calculated environmental dose rates for all analysed samples including loess standards as well as natural samples. For the sake of simplicity, we followed the suggestion proposed by Reviewer 1 to use assumed constant values for cosmic radiation ($0.150 \pm 0.015$ Gy/ka) and water content ($15 \pm 5\%$). Since these calculated dose rate values do not represent the actual environmental dose rates that might have been derived for the various locations when applying the actual water contents and cosmic radiations, the calculated dose rates are called 'simulated environmental dose rates' in the manuscript and in the plots. The details of dose rate calculation are described in the figure captions and the results are discussed in the manuscript.

*Aitken, M.J., 1985. Thermoluminescence dating. London. Academic Press.*

*Li, S.H., Tso, W.M., 1995. Systematic error from Th/U ratio in luminescence and ESR dating. Nuclear Science and Techniques 6, 113–116.*

- *To study the impact of measurement duration on the results, the authors have made repeated measurements with different durations. As the data can be stored during the measurements, I have a concern: are the short-measurements separate measurements or are the short measurements the former parts of long-measurements? I guess the former strategy is more reasonable, otherwise there would be correlation between the short and long-measurements.*

We agree with the reviewer that there would be a potentially problematic correlation between short-, medium- and long-time results if the short- and medium-time results were derived from only one long-lasting measurement. Therefore, all analysed measurements were performed as separate measurements. We did not use a long-lasting 'master'-measurement to derive different short-term parts. In order to clarify this aspect, we added an explanation to the manuscript.

- *When comparing the results of the 47 natural sediment samples, the discrepancy of several samples is attributed to the disequilibrium in U and Th decay chains. For example, fluvial flood plain sediments may have strongly alternating ground water levels which can increase or/and decrease specific radioactive daughter nuclides in the U and Th decay chains (line 483). In the beginning, I thought you meant the radon-loss induced disequilibrium, then I got confused. Because for the TSAC and gamma spectrometry measurements, the samples have been stored for 4 weeks before measurements and for μDose-system the samples have not been stored before measurements. If the discrepancy is caused by radon, the problem would exist for all samples. Now, I guess you meant the disequilibrium caused by long-lifetime daughter nuclides, right? Could you give examples of the daughter nuclides that might be influenced by the ground water level change, and if possible, list a reference? That may help the readers to better understand what you mean.*

The reviewer is right when assuming that we meant disequilibria caused by long-lifetime daughter nuclides. We added some sentences to the manuscript clarifying this point and illustrating the complex character of such potential disequilibria. We also added some references.

- *In the sample preparation step (line 195), the samples were pulverized in a ball mill (29.5 Hz for 45 minutes) and then dry sieved restricting the grain size diameter to < 63 μm. Do you assume that grain size < 63 μm would be fine enough for alpha counting? And usually how much sample would be left coarser than 63 um after being pulverized in a ball mill for 45 minutes? I am a little worried that this sieving step may cause fractionation of the sample component. For example, if the quartz is more difficult to grind than feldspar, the left residue of > 63 um will contain more quartz and the fine powder will have higher K (as well as Th, U) contents. Or maybe, the left residue of > 63 um contains more heavy minerals which have high Th, U contents (e.g. zircon), and the fine powder will have lower U, Th contents. Would it be better that we extend the grinding time and avoid the sieving process?*

We would like to thank the reviewer for this helpful comment pointing to a misleading verbalisation in the manuscript. When Reviewer 1 is asking "[…] how much sample would be left coarser than 63 μm after being pulverized in a ball mill for 45 minutes […]", I think he/she is most probably implying that a prolonged milling of 45 minutes will provide widely pulverized materials. And in fact, the reviewer is right in that assumption. The sieving step has been introduced in the Giessen Luminescence Laboratory as additional backstop for the sample preparation. If the amount of coarse-grained material is rather large (i.e., if there is any residual

> 63 µm) this indicates that the applied milling duration was not sufficient and that the whole sample (including the particles > 63 µm) should be subjected to further milling. Thus, the sieving step is not primarily intended to exclude grains with diameters > 63 µm, but to serve as an additional step to survey the quality of our preparation procedure. We added some additional sentences to the manuscript to clarify this point.

Up to now, residuals > 63 µm have been negligible for the utmost number of samples investigated in our laboratory when applying our long-lasting milling duration. In order to check this for the samples investigated in the submitted study I enquired our laboratory logbook in which our laboratory assistant records input (total sample) and output (< 63 µm) masses for all samples. Overall, only few samples so far investigated in Giessen showed suspicious results suggesting that the prolonged milling-procedure was not sufficient to pulverize the sample. None of these samples were used for the submitted study. As a result, we are confident that the impact of the sieving procedure can be regarded as negligible for the samples investigated in the submitted manuscript. This aspect was also amended to the manuscript.

Technical corrections:

*Line66: 'disk' is used here while 'disc' is used in line 197. Please make it consistent.*

Done. Everything was changed to 'disc'.

*Line79: 'These pairs are the result of…' change 'result' to 'results'.*

Done.

*Line174: '16.5 ± 1.5 mg/kg for K', change to 1.65 ± 0.15.*

Done. We changed the value to 1.65 ± 0.15 as well as the unit to % to make it consistent with the values used in the tables.

*Line 197: 'sample carrier', could you please indicate in Fig. 1a which one is sample carrier? And also give the name for that metal base.*

Done.

*Line 385: 'of only view hours', change 'view' to 'a few'.*

Done.

*Line 443: I think it is not necessary to use a separate paragraph here.*

Done.

*Table 4: Maybe it is better to also convert the activities of Murray et al. (2018) into the concentrations. That would give the readers a direct comparison of the results measured by different methods or labs.*

Done. We converted the activities provided by Murray et al. (2018) to concentrations. Additionally, all concentration values for which no activities were provided in the original publications were converted to Bq/kg. We used conversion factors provided by Guerrin et al (2011). All converted values are highlighted using a **-symbol which is explained in the table annotation. Additionally, we added a column showing 'simulated environmental dose rates' for each sample. Since IAG did not provide any information on the potassium content of the investigated Nussy standard, no dose rate was calculated for this dataset. The dose rates were calculated with DRAC (Durcan et al. 2015) based on the originally provided values for U, Th and K and assuming constant values for cosmic dose rates ($0.150 \pm 0.015$ Gy/ka) and water contents (15% $\pm$ 5 %).

---

## Author Comment (AC2)

**Reply to Reviewer 2**

We thank Reviewer 2 for his/her constructive comments and appreciate that he/she took the time to suggest some options to improve the submitted manuscript. We provide a detailed response to the raised concerns and suggestions below. For clarity, quotes from the review are italicized.

**Comments/Queries**

*Paragraph starting at L40: It's worth distinguishing here between emission counting and geochemical techniques, which may be relevant for considering data produced as part of the inter-laboratory comparison.*

Done. We rephrased the whole paragraph distinguishing between emission counting and geochemical techniques.

*L88: You mention secular equilibrium here, can you clarify whether the mDose can be used to identify disequilibrium? I guess not because the two U series are only sampled once, and your two Th pairs are fairly close together in the decay series.*

Reviewer 2 is right, when assuming that the µDose-system is in general not able to identify radioactive disequilibria. In fact, the specific algorithms used by the system require the sample to be either in or at least close to secular equilibrium. However, some types of radioactive disequilibria will return unusually high chi-squared values that might be used as indicators for such disequilibria.

*L104: Can you clarify please how repeated use of the standards will increase the accuracy of the calibration? To my mind, the accuracy (e.g. how well known the true value is) cannot be changed by repeated measurements, but precision perhaps can be monitored.*

We regret that the used verbalisation in the original version of the manuscript was not clear at this point. A central part of the calibration process is determining pair-specific calibration parameters that are used for the µDose-algorithms. These parameters are derived from a set of calibration measurements on material of known activities. The actual results obtained by these measurements are compared with the known values of the standards, which provides the base for the determination of the respective calibration parameters. Like other measurement procedures, the calibration measurements might be influenced by disturbing factors. Thus, the calibration is not only based on a single measurement of the respective standards, but on the mean of three repeated measurements of each standard providing a better estimate of the 'true values'. The more calibration measurements are considered for deriving the calibration parameters the less will these parameters be influenced by small random errors potentially affecting single measurements. We added some sentences to the manuscript in order to make the whole calibration procedure clearer.

*L105: Can you also clarify what you mean about the repeated measurements of the calibration? You mention later on in the manuscript that the calibration is repeated approximately every 6-8 months – is this what you mean here, or are you suggesting that the standard should be run after a fixed number of samples?*

In fact, in this paragraph (L99 – L107) we intended to express that the currently used calibration is not only based on a single measurement of the IAEA standards, but that it is always derived

from the means of three separate measurements which are performed for each of the three standards. Thus, the respective calibration is defined by the mean of three separate measurements of the RGU-1, the mean of three separate measurements of RGTh-1 and the mean of three separate measurements of RGK-1 as well as on one prolonged background measurement. Since IAEA standards are always measured for ~24 h and the background measurements comprise a duration of ~7 days, the whole calibration procedure requires a total of ~ 2 weeks.

This whole calibration procedure comprising 10 separate measurements (3x3 IAEA standards + 1 background) should be repeated at regular intervals. In the Giessen Luminescence Laboratory the devices are re-calibrated approximately every 6-8 months. We added some sentences to the manuscript to clarify this part.

*Sections 3.2 and 3.3: In these sections when reporting the various determined values, there is a switch between mg.kg-1 and Bq.kg-1. These come from the original publications, but it doesn't make it easy for the reader to navigate. It would be very helpful to convert one unit to the other, and this could be included table 4. Also, you present a number of different measurement datasets, but don't provide a rationale for why you chose the dataset that you did – is it simply a case of going for the first published values, or is there a different rationale?*

Done. We converted the values provided in the original publications to either concentrations or activities and compiled the information in Table 4. All converted values are highlighted using a **-symbol, which is explained in the table notes.

The different datasets providing measured values for the standards are inter alia intended to illustrate the order of variance associated with the standards. We are aware that the reference values reported in the cited publications vary much more than expected from the reported means and uncertainties. We chose the datasets of Preusser & Kasper (2001) and De Corte et al. (2007) since they are the datasets commonly cited in literature with respect to Nussy and Volkegem standards.

*Table 4: I think it would be really helpful here to provide a dose rate for these samples. I can see from the concentrations of U/Th/K that dose rates will be high, but off the top of my head, I don't know how high – this will provide context for later measurement time experiments.*

Done. We added a further column to the table showing a 'simulated environmental dose rate' for the samples. These dose rates were calculated with DRAC (Durcan et al. 2015) assuming unique water contents (15 ± 5%) and cosmic dose rates (0.150 ± 0.015 Gy/ka) for all samples. Thus, the calculated dose rate values do not correspond to the 'real' dose rates that might have been calculated for Nussy and Volkegem if the actual water contents and cosmic dose rates had been considered. In order to point out this fact, the term 'simulated environmental dose rate' is used.

*Table 5: I suggest strongly to move this table to the appendix/SI. It's two pages of data which isn't necessary for the paper and interrupts the flow. For completedness and good reporting practice, can the sampling location data for Heidelberg and Gliwice be included?*

Done. We moved the table to the appendix. The exact sampling locations for the Heidelberg and Gliwice samples have been included.

*L195: Was the sieving of the samples necessary after 45 minutes of milling? If you're filtering out sand size particles, it would be preferable to extend the milling time rather than sieve away these resilient mineral grains because you may be introducing bias.*

We would like to thank Reviewer 2 for this helpful comment. In fact, the verbalisation used in the manuscript is somehow misleading and can be interpreted in the sense that we filtered out sand sized particles, which would have caused a potentially problematic fractionation of mineral components.

When the reviewer is asking whether '[…] the sieving of the samples [was] necessary after 45 minutes of milling […]' I suppose that he/she might be implying that it was not. In fact, the reviewer is perfectly right in that assumption. With particular respect to the samples investigated in the submitted manuscript, we were not able to detect any residuals > 63 µm when applying the sieving procedure. From that point of view, the sieving step can be regarded as unnecessary.

However, we decided to keep the sieving step and mention it in the manuscript since it is part of our preparation procedure routinely applied in the Giessen Luminescence Laboratory. The sieving procedure was introduced as an additional step to survey the quality of the milling process. In general, we consider our extended milling duration as sufficient to provide widely pulverized material. However, long lasting practical experiences showed that this assumption is not always justified. In some exceptional cases even a prolonged milling duration might end up in samples still revealing substantial amounts of coarse grain material. This can either be explained by the sample-specific mineralogical composition (i.e. unusually high amounts of resistant minerals) or by user-specific peculiarities, such as filling the grinding beaker with too much material.

The sieving step provides a simple measure for identifying such insufficient milling procedures. If sand-sized particles with diameters of > 63 µm are detected, they are interpreted as indicator for an incomplete pulverization. In such a case, the sand-sized particles are not excluded from the sample, but the whole sample (including the residuals > 63 µm) is subjected to further milling. Reviewer 2 is perfectly right when he/she points out that excluding such resilient mineral grains would introduce bias to the measurement results. We added some sentences to the manuscript to explain the role of the sieving step in more detail.

As mentioned above, we were not able to detect any residuals > 63 µm for the samples investigated in the submitted study. Thus, we are confident that there was no significant impact of the sieving step and that we were not introducing any bias by applying the sieving procedure.

*L231: do you mean for samples with average dose rates?*

Yes, we wanted to express that measurement durations of ~2 – 4 days can be expected for samples with average dose rates that are typically reported for natural samples. We corrected the sentence in the manuscript in order to make it clearer.

*L248: I advise to change 'acceptable' to 'desirable' in this line – longer measurement times cannot be used as an excuse for not measuring a sample.*

Done.

*Section 4.3.3: this section is long and is passive in terms of not offering any value to the paper. Tabulating it would offer an easier means of digestion for the reader (e.g. columns for homogenisation techniques and determination of the radionuclides and/or dose rates). Alternatively, move this section to the appendix/SI.*

In fact, we believe this section to be an essential part of the manuscript as it provides the base information on the sample preparation and measurement procedures applied in the participating laboratories. We agree with the reviewer that a table briefly summarizing the basic information might be helpful. Thus, we added such a table to the manuscript. The detailed description of preparation and measurement procedures is now provided in the appendix.

*L321/325/334/351 (and potentially elsewhere): please avoid the use of excellent when making reference to accuracy/precision/results/reproducibility – it's subjective, descriptive, and meaningless. E.g. what is excellent accuracy? You're better off letting the data speak for itself.*

We tried to avoid the use of 'excellent'.

*Figure 2: can you offer some further explanation for the data in figure 2 please? When you say repeated measurement, do you mean literally repeat measurements on the same sub-sample, or do you mean that you re-sample the standard and re-measure?*

The term 'repeated measurements' means literally repeated measurements on the same subsample. So, there was no re-sampling. For each of the three IAEA standards, one 3g subsample was prepared for measurement. Once stored in the measurement container, the subsample was not removed from the container until all measurements on the respective device were completed. The same subsamples were used for all three devices.

*Clearly the K is reproducible, but there's more variability in the Th and the U. Is this due to heterogeneity in the sample if you're resampling each time, or is this a reflection of your measurement uncertainty?*

Since we used the same subsample for all measurements, the variability in the Th and U values cannot be attributed to the heterogeneity of different subsamples that might have been a reasonable explanation if we were re-sampling each time. Thus, variability in the Th and U values represent measurement uncertainties.

*Is there a reason that the Th shows more variability than the other two radionuclides?*

In fact, we do not have a final answer to this question. However, a possible explanation might be derived from Table 2 and 3 summarizing the reference values certified for the IAEA standards. Each standard is clearly dominated by one prominent component: RGK-1 by potassium (44.8%), RGTh-1 by thorium (800 ppm) and RGU-1 by uranium (400 ppm). The other components show small or negligible concentrations/activities. For RGK-1 and RGU-1, the concentrations of the non-dominant components are given as 'less than'-values indicating that the concentrations of these components are either close to the limit of detection or regarded as negligible. This is reflected by the respective activities given as NA-values for these components. In contrast, there are small but still detectable amounts of uranium (6.3 ppm) and potassium (0.02%) detected in RGTh-1. Since the detection of U- and Th-contents with the µDose-system is not independent of each other, these relatively high (compared with RGU-1

and RGK-1) portions of non-dominant components might be the reason for the slightly higher variability in the measured Th-values for RGTh-1.

*Section starting at L335: I agree with your conclusion at the end of page 17 (L351/2) that the IAEA measurements suggest the µDose is reproducible, and don't think that at first glance there is a problem with µDose (L346). However, there is a deviation of your results for Nussy and Vokegem loess standards and the published values you've chosen for comparison, and this should be further considered. Did you resample and remeasure your standards to assess intra-sample variability? This should be the first step. Then you can consider whether the discrepancies are due to 'methodological problems' (L349) associated with other techniques, and why these discrepancies might exist. It's not enough to state that 'more than 10% are neither unusual for dosimetry measurements ...' (L350), especially without any references (see works by Hossain et al., 2002, De Corte et al., 2007, Williams et al., 2010). Once you've considered the sources of these discrepancies, placing the deviations in the context of calculated dose rates would be a positive way of reassuring the reader that you're talking about minor absolute variations in U/Th, and that these don't have a significant impact on overall environmental dose rate, and therefore, age calculation.*

We agree with the reviewer that the results obtained for the loess standards Nussy and Volkegem reveal a deviation that needs to be discussed. The whole section starting at L335 is intended to provide such a discussion. In fact, we re-sampled the standards and re-measured them. These new results showed similar deviation as those mentioned in the manuscript. We added some sentences referring to these additional results. Following the suggestion of both reviewers, we calculated 'simulated environmental dose rates' and added a figure showing these dose rates together with bulk uranium and thorium activities. These additional findings are discussed in the manuscript. In the end, the dose rates based on the µDose-measurements of the Volkegem loess standard are in rather good agreement with the reference dose rate derived from the originally published values of De Corte et al. (2007). For the Nussy loess standard the results are less satisfying with respect to the reference values of Preusser & Kasper (2001). But still, they are within the order of magnitude reported by the IAG values and the values provided by Murray et al. (2018).

*Section 5.2: This is a really interesting section, and likely very useful for µDose users. As a general comment, I'm a bit lost understanding how the number of counts relates to i) time and ii) dose rate. Of course, these are sample specific, but for example knowing the dose rate of the loess standards and how long it took to reach 1k, 2k, 3k counts (for example) would be very helpful to the reader.*

In fact, some readers will be interested in this information. So, we provided an additional table for both loess standards in the Appendix. However, we would like to indicate that the informative value of this table is only very limited. The time necessary to reach a particular alpha count level will not only depend on the dose rate of the analysed sample, but also on the sample-specific composition of radionuclides. A sample revealing a low dose rate due to low K-40 activity may still have high uranium and thorium contents. Such a sample can reach the 3,000 alpha-count level much faster than a high dose rate sample with extremely high K-content but very low U- and Th-concentrations.

*L379: You start to explain the experiment here, but I don't quite understand how these measurement duration experiments were undertaken. Were you making only one long measurement (up to 5/7k counts) and integrating counts for the shorter count times, or were*

*you making numerous measurements (e.g. 0-250 cts, then 0-500, 0-1000 etc)? The former is preferable in my view.*

We performed numerous measurements on the same subsamples in order to avoid unwanted correlations that should have been expected if short- and medium-time values had been derived from the same 'master-measurement'. Thus, short-time, medium-time and long-time measurements were separate measurements and their results were not only derived from a long-lasting 'master-measurement'. Of course, the strategy of only conducting one long-lasting 'master-measurement' from which different time intervals are integrated would have had some advantages. Apart from a significant reduction of the total measurement time, this strategy would have avoided additional bias potentially introduced for instance by re-sampling procedures or by the loss of sample material when removing the subsamples from the devices. However, since all measurements were performed on the same subsamples of Nussy and Volkegem we did not have to apply any re-sampling. Furthermore, the used subsamples were not removed from the respective device until all required measurements were accomplished. Thus, we are confident that there was no significant loss of material.

*Paragraph starting L477: can you say more please about suspected disequilbirum issues, drawing from the literature about how disequilibrium might manifest in fluvial sediments.*

Done. We added some additional explanations and references illustrating the complex nature of radioactive disequilibria in environmental settings in general and with particular respect to fluvial sediments.

*You have pretty high Th values for these samples. Olley et al.'s 1996 study finds that U238 disequilibria is more common for their modern fluvial samples than U235 or Th. For all the samples you mention, Th is higher from the µDose system than Giessen TSAC/ICPOES – why would this be?*

The reviewer is right, when pointing out that the study of Olley et al. (1996) indicates that U-238 disequilibria are of greater importance for their fluvial samples than equilibria affecting the decay chain of Th-232. This is a general feature of natural samples and can be attributed to the relatively short lifetimes of daughter nuclides within the decay chain of Th-232 (e.g., Olley et al. 1997; Zöller & Schmidt 2020). As a result, secular equilibrium in the Th-232 decay chain will be re-established within a relatively short period of time (~ 30-40 years) after the system becomes closed again (e.g., Marsac et al. 2016; Zöller & Schmidt 2020). Thus, disequilibria in the Th-232 decay chain are often regarded as minor issue in sediment dating, at least when compared to disequilibria affecting the U-238 decay chain (Olley et al. 1996; Zöller & Schmidt 2020).

With radionuclide concentrations of > 20 ppm up to ~30 ppm, the Th-values measured for the samples potentially affected by disequilibria are indeed 'pretty high' in our study. This is true for both TSAC/ICP-OES measurements and µDose-measurements. However, we would like to indicate that there is no direct relationship between the importance/frequency of disequilibria in the Th-232 decay chain and the determined Th-values. If a sample is affected by radioactive disequilibria, a central assumption of the µDose-algorithms is violated. Independent of the respective nature of this radioactive imbalance, the calculated values based on µDose-measurements will most probably not be correct and therefore not agree with values derived from other methods.

Due to the complex nature of radioactive disequilibria it's hardly possible to give a general prediction on the specific direction or the extent of the resulting anomaly in U- and/or Th-concentrations. For the mentioned samples, the concentrations of Th are higher than usually expected for natural samples. This applies to the µDose-measurements as well as to the TSAC/ICP-OES measurements indicating that the potential disequilibria are affecting both methods, however, to different extents. This can be explained by the different ways (used pairs, specific algorithms) in which TSAC/ICP-OES and µDose-measurements are determining Th-concentrations.

The Th-values derived from the µDose-measurements are consistently higher than the values derived from TSAC/ICP-OES. This appears to be plausible when considering that all mentioned samples originate from similar environmental settings (fluvial deposits showing distinct features of fluctuating ground water levels). These similar environmental settings might probably show similar translocation processes. Thus, it seems possible and even quite likely that potential radioactive imbalances will be similar for the sediments which the mentioned samples were taken from and that these similar imbalances will end up in similar discrepancies between the applied methods.

Furthermore, it seems possible that some increase of Th-values is induced by Rn-220 emanation from the sample material. Unlike Rn-222, Rn-220 has a short half-life of only 55.6 s. When escaping from the sample it might decay before reaching the Rn-absorber inside the µDose-container. If this happens in front of the scintillator this can create additional decay pairs that might have some impact on the decay pair counts used for the determination of Th- and U-contents. However, the effect of this potentially disturbing factor is hard to study.

*Also, can you comment on the data for sample Gi455, where the Giessen and your data diverges for all three radionuclides. K won't suffer from equilibria issues, yet your measurements is 4 times greater than the Giessen one. Is this an experimental issue?*

The reviewer is right, when he/she is pointing to the fact that potassium should not suffer from equilibria issues in nature. And he/she is also right when supposing that the discrepancy between ICP-OES and µDose-measurements is caused by an experimental/methodological issue. The K-value determined by the µDose-system will be affected by radioactive disequilibria in the uranium and thorium decay chains since the activity arising from K-40 is not independently detected, but calculated as residual. The particular K-40 activity is derived from the excess of observed total beta-counts over the beta-counts expected to arise from the determined U-238, U-235 and Th-232 decay series. So, the K-40 activity will only be correct if the beta-counts for the uranium and thorium decay series are determined correctly. If a sample is affected by radioactive disequilibria, basic assumptions of the µDose-algorithms are violated. As a result, uranium and thorium concentrations won't be determined correctly. If the concentrations of U and Th are incorrect, neither the number of beta-counts arising from their respective decay series nor the residual beta-counts attributed to the decay of K-40 will be correct. In the end, the activity of K-40 will also be incorrect. This might be an explanation for the extraordinary deviation in the K-values of ICP-OES and µDose for sample Gi455.

*Why do you think there's such good agreement for this sample when you look at the comparison for HRGS in Figure 8b? Please offer a fuller explanation at L506 when you return to this issue*

In fact, the inconsistent results for sample Gi455 are really hard to explain. In the end, we will only be able to provide some possible explanations without being able to decide which might be the correct one.

When only considering the comparison of TSAC/ICP-OES and µDose, the results might be explained by radioactive disequilibrium, affecting the different methods to different extents (see above and discussion in the manuscript). In such a case, we would also expect similar discrepancies for the comparison between µDose and other methods such as HRGS. In fact, for Gi311 and Gi360, which were also supposed to suffer from radioactive disequilibria, such discrepancies can be observed. Yet for Gi455, HRGS and µDose show rather good agreement.

In the manuscript, we mentioned that '[t]his finding casts doubt on the above suggested explanation that Gi455 might suffer from a distinct radioactive disequilibrium'. In fact, the extraordinary large discrepancies observed for the TSAC/ICP-OES comparison and the good agreement of µDose and HRGS might indicate that something went wrong during the TSAC/ICP-OES measurements and therefore point to a measurement error of substantial extent. Particularly the amount of discrepancy observed for Gi455 is supporting this interpretation since other samples originating from the same sampling location (Gi450-Gi453) do not show similar discrepancies.

From a sedimentological point of view, sample Gi455 originates from floodplain loams, whereas samples Gi450-Gi453 were taken from gravel deposits. Based on our experience with sediments from the Lahn catchment, floodplain sediments of the region show significant higher concentrations of Th and U when compared to fluvial gravels. However, the TSAC/ICP-OES results obtained for Gi455 are in the same order of magnitude as the results obtained for Gi450-Gi453, which might be seen as an additional evidence suggesting that the TSAC/ICP-OES measurements of Gi455 were affected by serious problems.

However, in the end we cannot decide whether this was really the case. Radioactive disequilibrium still seems a possible explanation. As can be seen from the discussion of samples Col_UGW1 to Col_UGW4, for which we have strong evidence for radioactive imbalances, such disequilibria are not necessarily reflected by pronounced discrepancies between µDose-results and HRGS-results.

We added some sentences to the manuscript to give a short summary of possible reasons for the inconsistent results obtained for sample Gi455.

*Paragraph starting at 525: I suggest most strongly that you delete this paragraph, it's not correct. Dose rates calculated in the context of disequilbria are not accurate for trapped charge dating, no matter how precisely you can determine radionuclide concentrations, because ionisation is assumed to be constant through time. You haven't discussed how COL-UGW1-4 samples have been identified as being in disequilibrium or discussed the extent to which this results in excess/loss across the U/Th chains and how this might impact final dose rate calc.*

*L533: "At the moment, we cannot decide whether our preliminary results are only an odd anomaly or an indicator for the µDose-system's capability to produce reliable dose rate estimates even for samples suffering from radioactive disequilibria". µDose is a measurement tool for determining radioactivity. As far as I'm aware, it cannot be used to identify disequilibria in the U/Th decay series, and even if it were, there's no way of knowing of reconstructing the radioactivity history of a sample throughout it's burial history.*

*L534: "In order to give a final answer, further detailed and systematic investigations are required, including the question whether the magnitude of radioactive disequilibria is a decisive factor for the µDose-system's capability to determine correct values". I am entirely convinced that µDose can accurately and precisely determine U/Th/K and therefore infinite matrix dose rates. However, these infinite matrix dose rates rely on two fundamental assumptions i) of equilibrium and ii) of an infinite matrix.*

We would like to thank Reviewer 2 for pointing to this misleading passage in the manuscript. In fact, the reviewer is perfectly right when he/she is indicating that dose rates calculated from radionuclide concentrations of samples for which radioactive disequilibria have to be assumed will never be accurate for trapped charge dating.

Within the criticized paragraph (L524 – L536) we were actually using phrases such as 'correct determination' or 'produce reliable dose rate estimates'. These misleading verbalisations can indeed be interpreted in the sense that µDose-results might represent 'true' dose rate values even under conditions of radioactive imbalance. Such an interpretation would suggest that the µDose-system was able to identify radioactive disequilibria and, moreover, correct for them. However, we would like to stress that this is not the case. Consequently, the reviewer's comments on L533 and L534 are absolutely correct.

Therefore, we strongly regret having used such misleading formulations in this paragraph. In fact, the whole section was intended to deal with the rates of agreement between µDose-measurements and other applied methods. This also applies to the criticized paragraph, which is referring to samples from the Cologne Luminescence Laboratory for which we had strong evidence pointing to the presence of radioactive disequilibria. For such samples, we expected the µDose-measurements not to provide reliable results as the algorithms used for deriving radionuclide concentrations require secular equilibrium. Since determination of radionuclide activities in HRGS and µDose are based on differing approaches we expected large discrepancies between the applied methods for samples affected by radioactive disequilibria.

Although such discrepancies were detected for some of the analysed natural samples, this did obviously not apply to samples COL_UGW1 to COL_UGW4. When using verbalisations such as 'correct results' or 'reliable estimates' we actually wanted to use these phrases with respect to the particular benchmarks derived from HRGS. These phrases were intended to express that the determined µDose-results showed good agreement with results based on HRGS.

As a result, the reviewer's important and helpful criticism prompted us to revise the paragraph and to replace the misleading verbalisations.

*Table10 and the paragraph starting at L537: I personally don't think this table or section is relevant. Why wouldn't µDose be able to handle samples from a broad range of depositional settings? The problem only comes when the two dose rate assumptions mentioned above cannot be applied.*

Indeed, the reviewer is right when stating that the µDose-system should be able to handle samples from a broad range of depositional settings and that problems may only arise when the basic assumptions of secular equilibrium and infinite matrix are not applicable for the sediments from which the respective samples were collected. However, when presenting the preliminary results of our measurements at various conferences colleagues regularly asked whether we were able to identify specific environmental conditions for which the performance of the µDose-system was less satisfying. Of course, these colleagues regularly referred to the problem of

radioactive disequilibria. However, based on the well-known finding that specific types of sediment (i.e. fluvial deposits, soils, sediments enriched with organic matter etc.) are often regarded as prone to radioactive imbalances (e.g., Degering & Degering, 2020), the discussions were often focussing on these specific environments in general and not on the specific issue of radioactive imbalance.

Thus, we would like to keep Table 10 and the paragraph starting at L537 as an additional piece of information. We think that it might particularly be helpful for readers whose working focus is not in the field of dosimetry but rather in the field of applied dating approaches. The results presented in Table 10 and discussed in the text indicate that µDose-results are comparable with results derived from well-established methods of environmental dosimetry. Based on the summary in Table 10 we can conclude that there is no environmental/depositional setting that is obviously not suitable for µDose-measurements and should therefore generally be avoided. Such concerns were, for instance, raised by some colleagues with respect to fluvial sediments.

However, we think that the critical objection raised by Reviewer 2 should be considered in this paragraph since the reliability of obtained µDose-results always depends on the mentioned assumptions of secular equilibrium and infinite matrix. Therefore, we amended some sentences to the manuscript considering that fact.

**Minor corrections/typos**

*L23: hyphenation not required, instead "heating events, and for ESR dating, the precipitation of minerals"*

Done.

*L30: replace "minerals are not stimulated any more" with "minerals are no longer stimulated"*

Done.

*L30: replace "are still" with "remain"*

Done.

*L31: remove comma midway through the sentence*

Done.

*L32: insert "the" in front of palaeodose*

Done.

*L44: replace "as well as" with "additionally"*

We revised the whole paragraph.

*L51: remove "here"*

Done.

*L58: I think you mean detection rather than determination*

The reviewer is right. The term was changed.

*L71: Rewrite sentence to "..an Analogue to Digital Converter (ADC) samples and transforms the …"*

Done.

*L74: Replace "allowing to discriminate" with "allowing discrimination"*

Done.

*L75: Rewrite sentence to: "..pulses, as well as the elimination of background pulses caused by interfering variables"*

Done.

*L103: Replace "reveal rather" with "have"*

Done.

*L204: Replace "might have a tampering effect on" with "may impact"*

Done.

*L365: Correct 'acitivity' typo*

Done.

*Table 7: is "1.04 ± 0.003" a typo for the Nussy loess K% on Ahnert value?*

In fact, a typo. We changed the whole table. All errors for the µDose-measurements are now given as 95% C.I.s, which will provide a better comparability to the reference values given in the table.

*L368/9: replace 'rather fast' with 'relatively rapid'*

Done.

*L369/70: amend sentence to "without the need for storage for specific periods of time"*

Done.

*L371: do you mean precision instead of quality here?*

We used 'quality' as a general term.

*L385: replace "view" with "a few"*

Done.

*L557/558: should be one paragraph*

Done.

*L565: these pieces of software should be referenced*

Done.

**Citation**: https://doi.org/10.5194/gchron-2021-20-RC2